# Revisiting stochastic submodular maximization with cardinality constraint: A bandit perspective

**Pratik Jawanpuria**                                          *pratik.jawanpuria@microsoft.com*
*Microsoft, India*

**Bamdev Mishra**                                                   *bamdevm@microsoft.com*
*Microsoft, India*

**Karthik S. Gurumoorthy**                              *karthik.gurumoorthy@walmart.com*
*Walmart Global Tech, India*

**Reviewed on OpenReview:** *https://openreview.net/forum?id=57ETChLAOE*

## Abstract

In this paper, we focus on the problem of maximizing non-negative, monotone, stochastic submodular functions under cardinality constraint. Recent works have explored continuous optimization algorithms via multi-linear extensions for such problems and provided appropriate approximation guarantees. We take a fresh look into this problem from a discrete, (stochastic) greedy perspective under a probably approximately correct (PAC) setting, i.e., the goal is to obtain solutions whose expected objective value is greater than or equal to $(1 - 1/e - \epsilon)\mathrm{OPT} - \nu$ with at least $1 - \delta$ probability, where OPT is the optimal objective value. Using the theory of multi-armed bandits, we propose novel bandit stochastic greedy (BSG) algorithms in which selection of the next element at iteration $i$ is posed as a $(\nu_i, \delta_i)$-PAC best-arm identification problem. Given $(\nu, \delta)$-PAC parameters to BSG, we formally characterize a set $\mathcal{A}(\nu, \delta)$ of per-iteration policies such that any policy from this set guarantees a $(\nu, \delta)$-PAC solution for the stochastic submodular maximization problem using BSG. We next discuss the problem of learning a policy in $\mathcal{A}(\nu, \delta)$ by minimizing the computational cost. With our learned policy, we show that BSG has lower computational cost than existing stochastic submodular maximization approaches. An interesting outcome of our analysis is the development of both linear and almost-linear time algorithms for the exemplar based clustering problem with $(1 - 1/e - \epsilon)$-approximation guarantee under a PAC setting. Lastly, we also study the problem of learning a policy for BSG under budget setting. Experiments on various problems illustrate the efficacy of our approach in terms of optimization quality as well as computational efficiency.

## 1 Introduction

Submodular functions are those that have the property of diminishing returns, where the incremental gain of adding an element to a set decreases as the size of the set increases. They are often regarded as discrete analogs of convex functions and arise naturally in applications such as document summarization (Bairi et al., 2015; Kim et al., 2016; Gurumoorthy et al., 2019), image summarization (Tschiatschek et al., 2014), data/feature subset selection (Wei et al., 2015), speech data training (Wei et al., 2014), facility location (Mehrotra & Vishnoi, 2023; Karimi et al., 2017; Agarwal et al., 2023), influence maximization in social networks (Kempe et al., 2003; Chen et al., 2009), sensor placement, active learning (Wei et al., 2014), and recommender systems (Mehrotra & Vishnoi, 2023), to name a few. We refer interested readers to the survey (Krause & Golovin, 2014) for further introduction to submodular functions.

Given a ground set $\mathcal{S} = [n]$ consisting of first $n$ natural numbers, a set function $F : 2^{\mathcal{S}} \to \mathbb{R}$ is submodular if it satisfies $F(A) + F(B) \geq F(A \cap B) + F(A \cup B)$ for all $A, B \subseteq \mathcal{S}$. Defining $g(i|A) = F(A \cup \{i\}) - F(A)$

as the incremental gain of adding an element $i$ to the set $A$, a submodular function equivalently satisfies the diminishing returns property: $g(i|A) \geq g(i|B)$ for any $A \subseteq B \subseteq \mathcal{S}$ and $i \notin B$. The function is monotone non-decreasing if $g(i|A) \geq 0, \forall i \in \mathcal{S} \setminus A$, and non-negative when $F(A) \geq 0, \forall A \subseteq \mathcal{S}$. The submodular maximization problem is typically stated as identifying the subset $\mathbf{s}^*$ under some constraints that maximizes the submodular function $F$. In this regard, a common optimization problem is maximizing $F$ under a cardinality constraint, stated as:

$$\max_{\mathbf{s} \subseteq \mathcal{S}, |\mathbf{s}| \leq k} F(\mathbf{s}). \tag{1}$$

As submodular maximization is NP-hard in general, fast algorithms are sought to approximate $\text{OPT} \coloneqq \max_{\mathbf{s} \subseteq \mathcal{S}, |\mathbf{s}| \leq k} F(\mathbf{s})$ with theoretical guarantees (Nemhauser et al., 1978).

Problem (1) is classically studied in an oracle model, where the access to $F$ is made available via a black box which when fed a subset $\mathbf{s}$ outputs $F(\mathbf{s})$. However, recent works (Karimi et al., 2017; Mokhtari et al., 2018) have also investigated (1) in settings where $F$ is estimated from large amount of data involving stochastic fluctuations, or is not directly accessible and computed only via simulations. Examples of such applications include influence maximization, exemplar-based clustering, recommender systems, prototype selection, etc. In influence maximization, for instance, the goal is to find the most influential seed nodes (of size $k$) that maximize the propagation of influence through the network (Kempe et al., 2003). The subset of nodes influenced from a seed set is modeled as an expectation of a stochastic process, and may not be computable in closed-form. In facility location, the objective may be viewed as an expected value of similarity between a large number of data points and chosen $k$ centers (with respect to the distribution of data points).

In the above applications, the objective function for stochastic submodular maximization (SSM) problems could be expressed as $F(\mathbf{s}) \coloneqq \mathbb{E}_{\mathbf{z} \sim P}[f(\mathbf{s}; \mathbf{z})]$, where $f$ is a stochastic function and $\mathbf{z} \in \mathcal{Z}$ is a random variable with distribution $P$. When $F(\cdot)$ is computed from large number of data points or simulations, standard discrete algorithms such as (lazy) Greedy (Minoux, 1978) or (lazy) Stochastic Greedy (Mirzasoleiman et al., 2015) incur a high computational cost since and require calling the stochastic function $f$ large number of times for each input subset $\mathbf{s}$. Recent works (Karimi et al., 2017; Hassani et al., 2017; Mokhtari et al., 2018; 2020) have proposed stochastic continuous optimization algorithms for SSM which leverage the multi-linear extension (Vondrák, 2008; Calinescu et al., 2011) of submodular problems.

In this work, we take a fresh look into non-negative monotone SSM problems with cardinality constraint from a greedy perspective and under provably approximately correct (PAC) setting. In each iteration of the (stochastic) greedy approach, we view the set of candidate elements as *arms* with expected rewards equaling their incremental gains. Hence, choosing the candidate with the highest incremental gain (highest reward) is similar to the best-arm identification problem in the multi-armed bandit (MAB) literature (Even-Dar et al., 2006; Audibert et al., 2010). Given $(\nu, \delta)$-PAC parameters, we first define a $(\nu, \delta)$-PAC solution (5) for the SSM problem (2). We next propose novel bandit stochastic greedy (BSG) algorithm for solving SSM problems with cardinality constraint. The next element selection problem in each iteration $i$ of BSG is posed as a $(\nu_i, \delta_i)$-PAC best-arm identification problem. The main contributions of this work is the analysis of BSG in the following directions.

**1** Correctness of BSG: given $(\nu, \delta)$-PAC parameters, can the proposed BSG algorithm obtain a $(\nu, \delta)$-PAC solution for the SSM problem (2)? We answer this affirmatively by characterizing a set $\mathcal{A}(\nu, \delta)$ of per-iteration policies which guarantee BSG to obtain a $(\nu, \delta)$-PAC solution. This is discussed in Section 3.2.

**2** Given $(\nu, \delta)$-PAC parameters for BSG, we note that the set $\mathcal{A}(\nu, \delta)$ has several $\{(\nu_i, \delta_i)\}_{i=1}^{k}$ per-iteration policies. An immediate question is: what is BSG's computational cost with such policies in terms of the number of stochastic function $f$ calls ($N$)? In Section 3.3, we discuss the computational cost of BSG for a given $\{(\nu_i, \delta_i)\}_{i=1}^{k}$ policy. We also investigate the problem of learning a policy $\{(\nu_i, \delta_i)\}_{i=1}^{k} \in \mathcal{A}(\nu, \delta)$ by minimizing the computational cost.

**3** Furthermore, we explore the problem of minimizing BSG's error parameter $\nu$ under a given fixed budget of $N_0$ stochastic function $f$ calls and a given confidence level $\delta$. In this setting, we learn a policy $\{(\nu_i, \delta/k)\}_{i=1}^{k} \in \mathcal{A}(\nu, \delta)$ for BSG such that the overall error level $\nu$ is minimized. This is discussed in Section 3.4.

Table 1: Approximation guarantees and number of stochastic function $f$ calls ($N$) for various algorithms on the SSM problem (2). Let $\gamma(\epsilon) = 1 - 1/e - \epsilon$ and $m$ denotes the maximum number of stochastic function $f$ calls required to compute the function $F(\cdot)$. In exemplar based clustering, for instance, $m = n$. For discrete stochastic methods (disc. stoch.), the expectation over $F(\mathbf{s})$ is over the randomized sampling of the candidate set. For continuous stochastic methods (cont. stoch.), the expectation over $F(\mathbf{s})$ is due to multi-linear extension, stochastic gradients, and randomized pipage rounding. Greedy is the only deterministic method in the table. We observe that the proposed algorithms (BG-ABA, BG-NE, BSG-ABA, and BSG-NE) have lower complexity of $N$ in terms of $n$.

| Algorithm | Type | Approximation guarantee | Number of $f$ calls ($N$) |
|---|---|---|---|
| Greedy | discrete | $F(\mathbf{s}) \geq \gamma(0)\mathrm{OPT}$ | $O(nkm)$ |
| BG-ABA (**this work**) | discrete | $\mathbb{P}[F(\mathbf{s}) \geq \gamma(0)\mathrm{OPT} - \nu] \geq 1 - \delta$ | $O(nk^3 \log(\frac{k}{\delta})/\nu^2)$ |
| BG-NE (**this work**) | discrete | $\mathbb{P}[F(\mathbf{s}) \geq \gamma(0)\mathrm{OPT} - \nu] \geq 1 - \delta$ | $O(nk^3 \log(\frac{nk}{\delta})/\nu^2)$ |
| Stochastic Greedy (Mirzasoleiman et al., 2015) | disc. stoch. | $\mathbb{E}[F(\mathbf{s})] \geq \gamma(\epsilon)\mathrm{OPT}$ | $O(nm \log(1/\epsilon))$ |
| BSG-ABA (**this work**) | disc. stoch. | $\mathbb{P}[\mathbb{E}[F(\mathbf{s})] \geq \gamma(\epsilon)\mathrm{OPT} - \nu] \geq 1 - \delta$ | $O(nk^2 \log(\frac{k}{\delta}) \log(\frac{1}{\epsilon})\gamma(\epsilon)^2/\nu^2)$ |
| BSG-NE (**this work**) | disc. stoch. | $\mathbb{P}[\mathbb{E}[F(\mathbf{s})] \geq \gamma(\epsilon)\mathrm{OPT} - \nu] \geq 1 - \delta$ | $O(nk^2 \log(\frac{n}{\delta} \log(\frac{1}{\epsilon})) \log(\frac{1}{\epsilon})\gamma(\epsilon)^2/\nu^2)$ |
| SGA (Hassani et al., 2017) | cont. stoch. | $\mathbb{E}[F(\mathbf{s})] \geq (1/2)\mathrm{OPT} - \nu$ | $O(n^2k^2/\nu^2)$ |
| SCG (Mokhtari et al., 2018) | cont. stoch. | $\mathbb{E}[F(\mathbf{s})] \geq \gamma(0)\mathrm{OPT} - \nu$ | $O(n^{2.5}k^{1.5}/\nu^3)$ |
| SCG++ (Hassani et al., 2019) | cont. stoch. | $\mathbb{E}[F(\mathbf{s})] \geq \gamma(0)\mathrm{OPT} - \nu$ | $O(n^2k^4/\nu^2)$ |

In addition to the BSG algorithm, we also propose a (non-randomized) variant of BSG, henceforth termed as bandit greedy (BG) algorithm. In BG, all candidate arms are considered in every iteration unlike BSG where only a few candidate arms are randomly chosen per iteration. Hence BG has a higher computation cost than BSG, but better approximation guarantee. The approximation guarantees and the computational costs of the proposed the BG and BSG algorithms with our learned policies are summarized in Table 1.

In Table 1, we observe that our proposed algorithms have either linear or almost-linear computational cost in $n$. An interesting implication of our result is that the proposed algorithms reduces the computational cost of each Greedy (or Stochastic Greedy) iteration for the exemplar based clustering (also known as $k$-medoids clustering) problem from quadratic to linear or almost-linear and provide similar approximation guarantee as Greedy (or Stochastic Greedy) under a PAC setting.

Empirical results illustrate the efficacy of the proposed approach on applications such as exemplar based clustering and representative sampling from a target set. The proofs of all theoretical results and additional experimental details are provided in the appendix.

## 2 Related work

This section briefly reviews the literature on submodular maximization, stochastic submodular maximization, and the best-arm identification (or pure exploration) problem under the multi-armed bandit framework.

### 2.1 Submodular maximization

**Discrete greedy approaches.** For maximizing a monotone, non-negative, deterministic submodular function $F$ under cardinality constraint $k$, the landmark paper (Nemhauser et al., 1978) employed the Greedy algorithm to obtain the $(1 - 1/e)$ approximation guarantee, and proved that any algorithm that is allowed to only evaluate $F$ at a polynomial number of sets will not be able to obtain an approximation guarantee better than $(1 - 1/e)$. Exploiting the properties of submodularity, Minoux (1978) proposed an accelerated version of the classical greedy algorithm, popularly known as Lazy Greedy (Leskovec et al., 2007). The Stochastic Greedy algorithm (Mirzasoleiman et al., 2015) provides $(1 - 1/e - \epsilon)$ approximation in expectation using

$O(n \log(\frac{1}{\epsilon}))$ function $F$ evaluations. A deterministic $(1 - 1/e - \epsilon)$ approximation algorithm requiring $O(n/\epsilon)$ calls to $F$ has been recently proposed (Li et al., 2022).

**Continuous greedy approaches.** Existing works (Vondrák, 2008; Calinescu et al., 2011; Chekuri et al., 2014) have explored continuous relaxations of submodular functions via multi-linear extension, which lifts the discrete problem (1) into a continuous domain and perform continuous optimization using gradients. Calinescu et al. (2011) showed that the continuous greedy algorithm obtains the tight $(1 - 1/e)$ approximation bound for monotone submodular functions under a general matroid constraint. Randomized pipage rounding (Calinescu et al., 2011; Karimi et al., 2017) is usually employed to obtain a discrete solution from fractional solution without worsening the objective value in expectation.

## 2.2 Stochastic submodular maximization (SSM)

While the classical Greedy or the Stochastic Greedy (Mirzasoleiman et al., 2015) algorithms may be employed for solving SSM problem ($F(\mathbf{s}) := \mathbb{E}_{\mathbf{z} \sim P}[f(\mathbf{s}; \mathbf{z})]$) defined over large number of data points or simulations, they compute the function $F(\cdot)$ exactly at each iteration $i$. Hence, they require the full data at each iteration and incur high computational cost. Recent works have explored stochastic gradient based continuous greedy approaches for SSM as they require only a small batch of data at every iteration.

For maximizing weighted coverage functions, a subclass of stochastic submodular functions, Karimi et al. (2017) proposed a concave relaxation scheme and employed projected stochastic gradient ascent (SGA) algorithm. However, for SSM problems, SGA does not provide tight guarantees as it offers $\text{OPT}/2 - \nu$ lower bound in expectation after $O(n^2 k^2/\nu^2)$ iterations (Hassani et al., 2017). Conditional gradient based stochastic continuous greedy (SCG) improves upon SGA and achieves $(1 - e^{-1})\text{OPT} - \nu$ guarantee in expectation after $O(n^{2.5} k^{1.5}/\nu^3)$ iterations (Mokhtari et al., 2020). Hassani et al. (2020) proposed variance reduction based SCG, SCG++, which provides the same guarantee as SCG in $O(n^2 k^4/\nu^2)$ iterations.

Table 1 summarizes the approximation guarantee and the computational cost (in terms of the number of stochastic function $f$ calls) corresponding to Greedy, Stochastic Greedy, the stochastic gradient based continuous greedy approaches (SGA, SCG, and SCG++), and our proposed algorithms.

## 2.3 Best-arm identification problem

Consider a set $\mathcal{S}$ of $n$ arms. When an arm $s \in \mathcal{S}$ is sampled (i.e., pulled), a reward $\mu(s)$ is received. This reward is realization of a random variable from an unknown distribution $D(s)$. The pure exploration (or best arm identification) problem (Bubeck et al., 2009; Audibert et al., 2010) is defined as identifying an arm $s^*$ with the highest expected reward: $s^* \in \arg\max_{s \in \mathcal{S}} \mathbb{E}_{D(s)}[\mu(s)]$. Thus, estimating the expected reward of an arm involves sampling them sufficiently. Learning a $(\nu, \delta)$-best arm under probably approximately correct (PAC) setting implies identifying an arm $s'$ such that $\mu(s') \geq \mu(s^*) - \nu$ with at least $1 - \delta$ probability. Here, $\nu$ and $\delta$ are usually termed as the error and confidence parameters, respectively. An interesting problem is to identify a $(\nu, \delta)$-best arm while incurring minimal sampling complexity (Jamieson & Nowak, 2014).

Elimination strategy is a common approach for the $(\nu, \delta)$-PAC problem, with median elimination (Even-Dar et al., 2006) being one of the most popular algorithms. Elimination framework involves sampling from all the non-eliminated arms at each around and eliminating some of the arms at the end of each round. The celebrated Hoeffding bound provides the following simple yet efficient naive elimination (NE) procedure for obtaining a $(\nu, \delta)$-best arm $s' \in \mathcal{S}$:

1. Given $\nu, \delta$ parameters, sample each arm $2 \log(n/\delta)/\nu^2$ times and record the mean empirical reward $\hat{\mu}(s)$ for each arm $s \in \mathcal{S}$

2. Select the arm $s' := \underset{s \in \mathcal{S}}{\arg\max} \hat{\mu}(s)$.

Recently, Hassidim et al. (2020) proposed the *approximate best arm* (ABA) algorithm that learns a $(\nu, \delta)$-best arm with lower number of samples than the median elimination algorithm. The ABA algorithm employs two

elimination strategies – aggressive elimination (AE) and NE – one after the other. The aim of AE is to use only a few samples to eliminate a significant number of potentially uninteresting arms with high confidence. The subsequent NE stage takes the surviving arms from AE as well as a randomly chosen subset of $\mathcal{S}$ (to account for the small probability that AE may eliminate good arms) and outputs a $(\nu, \delta)$-best arm. We detail the steps of the ABA algorithm in Section G.

Within the best-arm identification framework, another interesting setting is the fixed-budget setting (Audibert et al., 2010; Gabillon et al., 2012; Karnin et al., 2013; Jamieson & Nowak, 2014). Here, the aim is to maximize the confidence (i.e., provide the best guarantee) of selecting the best arm given a fixed number of sampling, i.e., arm pulls.

## 3 Proposed approach

Given a ground set $\mathcal{S} = \{1, \ldots, n\}$ of $n$ elements, we consider the problem of maximizing a non-negative, monotone, stochastic submodular function $F : 2^{\mathcal{S}} \to \mathbb{R}_+$, subject to cardinality constraint, in the setting where the function $F$ is defined as $F(\mathbf{s}) := \mathbb{E}_{\mathbf{z} \sim P}[f(\mathbf{s}; \mathbf{z})]$. The function $f : 2^{\mathcal{S}} \times \mathcal{Z} \to \mathbb{R}_+$ may be given as a stochastic function (as part of an implicit stochastic model for which the distribution $P$ may be unknown) or could be an empirical function (with possibly large domain $\mathcal{Z}$). Overall, the SSM problem is formalized as follows:

$$\max_{\mathbf{s} \subseteq \mathcal{S}, |\mathbf{s}| \leq k} F(\mathbf{s})(:= \mathbb{E}_{\mathbf{z} \sim P}[f(\mathbf{s}; \mathbf{z})]). \tag{2}$$

It should be noted that the stochastic function $f(\mathbf{s}; \mathbf{z})$ need not be submodular (Mokhtari et al., 2018; 2020). Since our analysis involves application of Hoeffding's inequality, we assume that the range of $f$ is $[0, 1]$ for simplified expressions. As with existing works on approximate best-arm identification problem (Bagaria et al., 2018; Hassidim et al., 2020), all theoretical results of this paper can be generalized to any sub-Gaussian distribution.

The classical Greedy algorithm (Nemhauser et al., 1978) for (2) begins with the empty set $\mathbf{s}_0 = \emptyset$. In the $i$-th iteration, it finds the element $s$ that maximizes the incremental gain $g_i$, i.e., $s \in \arg\max_{v \in \mathcal{R}_i} g_i(v|\mathbf{s}_{i-1})$, where $\mathcal{R}_i = \mathcal{S} \setminus \mathbf{s}_{i-1}$ is the candidate element set and $g_i(v|\mathbf{s}_{i-1}) = F(\mathbf{s}_{i-1} \cup \{v\}) - F(\mathbf{s}_{i-1})$ is the incremental gain as defined earlier. We then update $\mathbf{s}_i \leftarrow \mathbf{s}_{i-1} \cup \{s\}$. On the other hand, Stochastic Greedy (Mirzasoleiman et al., 2015) chooses a smaller (but random) candidate element set $\mathcal{R}_i \subseteq \mathcal{S} \setminus \mathbf{s}_{i-1}$ for the $i$-th iteration, with $|\mathcal{R}_i| = \frac{n}{k} \log(\frac{1}{\epsilon})$.

Both Greedy and Stochastic Greedy require multiple evaluations of the function $F(\cdot)$, which may be computationally costly or even impractical in several real-world applications (Mokhtari et al., 2018; 2020). We propose novel discrete stochastic greedy based algorithms that require a small number of stochastic function $f$ calls in each iteration and provide approximation guarantees.

### 3.1 Bandit stochastic greedy (BSG)

Each iteration of the stochastic greedy framework involves greedily selecting an element with the maximum incremental gain. The incremental gain for the SSM problem (2) with respect to an element $s \in \mathcal{R}_i$ may be written as:

$$\begin{aligned} g(s|\mathbf{s}) &= \mathbb{E}_{\mathbf{z} \sim P}[f(\mathbf{s} \cup \{s\}; \mathbf{z})] - \mathbb{E}_{\mathbf{z} \sim P}[f(\mathbf{s}; \mathbf{z})] \\ &= \mathbb{E}_{\mathbf{z} \sim P}[g(s|\mathbf{s}; \mathbf{z})], \end{aligned} \tag{3}$$

where $g(s|\mathbf{s}; \mathbf{z}) := f(\mathbf{s} \cup \{s\}; \mathbf{z}) - f(\mathbf{s}; \mathbf{z})$ is the stochastic marginal gain (corresponding to the sample $\mathbf{z} \sim P$) when an element $s$ is added to $\mathbf{s}$. Since (3) is an expectation, we may obtain an unbiased estimator of $g(s|\mathbf{s})$ as $\hat{g}(s|\mathbf{s}; \mathcal{Z}_i) = \sum_{\mathbf{z} \in \mathcal{Z}_i} g(s|\mathbf{s}; \mathbf{z})/|\mathcal{Z}_i|$ given a set $\mathcal{Z}_i$ whose elements are sampled from $P$. We also note that the term $\mathbb{E}_{\mathbf{z} \sim P}[f(\mathbf{s}; \mathbf{z})]$ in (3) is constant with respect to the new element $s$. Thus, we may equivalently use the estimator $\hat{h}(s|\mathbf{s}; \mathcal{Z}_i) = \sum_{\mathbf{z} \in \mathcal{Z}_i} f(\mathbf{s} \cup \{s\}; \mathbf{z})/|\mathcal{Z}_i|$ for greedily selecting the next element.

We now propose to employ the best arm selection strategies from the stochastic multi-armed bandit (MAB) framework for greedily selecting the next element. We view the different candidates $s \in \mathcal{R}_i$ as individual arms (agents) with the corresponding reward given by $\hat{h}(\cdot)$. Thus, identifying the best element that maximizes the

---

**Algorithm 1** Proposed bandit greedy (BG) and bandit stochastic greedy (BSG) algorithms

---

**Input:** $\mathcal{Z}, k, \nu, \delta$, and $\epsilon$.
**Select** a per-iteration $\{(\nu_i, \delta_i)\}_{i=1}^k$ policy satisfying (6).
Initialize $\mathbf{s}_0 = \phi, i = 1$.
**while** $i \leq k$ **do**
    **1: BG:** $\mathcal{R}_i = \mathcal{S} \setminus \mathbf{s}_{i-1}$
      or
    **BSG:** $\mathcal{R}_i = $ a subset of $\mathcal{S} \setminus \mathbf{s}_{i-1}$ with $nk^{-1}\log(1/\epsilon)$ elements which are selected uniformly at random.
    **2:** $s = \text{PAC\_BEST\_ARM}(\mathbf{s}_{i-1}, \mathcal{R}_i, \mathcal{Z}, \nu_i, \delta_i)$.
    **3:** $\mathbf{s}_i = \mathbf{s}_{i-1} \cup \{s\}, i = i + 1$.
**end while**
**Output:** $\mathbf{s}_k$.

---

incremental gain under $(\nu, \delta)$-PAC setting may be viewed as a pure exploration problem (Even-Dar et al., 2006; Bubeck et al., 2009). If $s_i^*$ is the element with best incremental gain (3) in the $i$-th iteration of the (stochastic) greedy framework, a $(\nu_i, \delta_i)$-best element $s_i$ satisfies the following condition:

$$\mathbb{P}[g(s_i|\mathbf{s}_{i-1}) \geq g(s^*|\mathbf{s}_{i-1}) - \nu_i] \geq 1 - \delta_i. \tag{4}$$

In Algorithm 1, we formalize both our bandit greedy (BG) and bandit stochastic greedy (BSG) approaches. The difference between BG and BSG approaches is only in Step 1, which is in the candidate element set $\mathcal{R}_i$. As observed in Step 1 of Algorithm 1, BG has the candidate element set $\mathcal{R}_i$ as the set of all the remaining elements ($\mathcal{R}_i = \mathcal{S} \setminus \mathbf{s}_{i-1}$). On the other hand, BSG selects the candidate element set as a random subset of the remaining elements with $|\mathcal{R}_i| = nk^{-1}\log(1/\epsilon)$, where $\epsilon$ is the stochastic greedy parameter (Mirzasoleiman et al., 2015). This randomly selected candidate set lowers the computational cost of BSG with respect BG by $O(k)$ (Table 1). The `PAC_BEST_ARM` function in Step 2 of Algorithm 1 denotes a best-arm identification algorithm that returns a $(\nu_i, \delta_i)$-best element at $i$-th iteration. The set $\{(\nu_i, \delta_i)\}_{i=1}^k$ of PAC parameters employed in Algorithm 1 is termed as a policy. Step 3 of Algorithm 1 updates the solution set $\mathbf{s}_i$ with the chosen element $s$ that is output of Step 2.

In this work, we analyze the proposed BG and BSG approaches with `PAC_BEST_ARM` as the NE or the ABA algorithm (discussed in Section 2.3). If ABA is employed in Algorithm 1, we term it BSG-ABA (or BG-ABA) and if NE is employed, we term it BSG-NE (or BG-NE). In the rest of this section, we discuss the approximation guarantees provided by our BSG algorithms. Analogous results for the BG algorithms are discussed in Section C. Table 1 summarizes the key theoretical results for the BSG-ABA, BSG-NE, BG-ABA, and BG-NE algorithms.

In the stochastic greedy framework with the (stochastic greedy) parameter $\epsilon$, a set $\mathbf{s} \subseteq \mathcal{S}$ (with $|\mathbf{s}| \leq k$) is termed as a $(\nu, \delta)$-PAC solution of (2) if the following holds:

$$\mathbb{P}\left(\mathbb{E}\left[F(\mathbf{s})\right] \geq (1 - 1/e - \epsilon)\text{OPT} - \nu\right) \geq 1 - \delta. \tag{5}$$

We next analyze the following questions related to Algorithm 1:

1. Given $(\nu, \delta)$-PAC parameters, can Algorithm 1 obtain a $(\nu, \delta)$-PAC solution for the SSM problem (2)? If yes, can we characterize a set $\mathcal{A}(\nu, \delta)$ of policies such that any policy from this set would ensure Algorithm 1 obtains a $(\nu, \delta)$-PAC solution for (2)?

2. What is the computational cost of Algorithm 1 for a given $\{(\nu_i, \delta_i)\}_{i=1}^k$ policy? Can we learn a policy $\{(\nu_i, \delta_i)\}_{i=1}^k \in \mathcal{A}(\nu, \delta)$ by minimizing the computational cost?

3. Can we learn a policy $\{(\nu_i, \delta_i)\}_{i=1}^k \in \mathcal{A}(\nu, \delta)$ by minimizing the overall error level $\nu$ corresponding to Algorithm 1 under a given budget of computational cost and a given confidence level $\delta$?

### 3.2 Characterizing a set of policies which guarantee BSG to learn a $(\nu, \delta)$-PAC solution

Our first result shows that the proposed BSG algorithm indeed learns a $(\nu, \delta)$-PAC solution of the SSM problem (2) if the $\{(\nu_i, \delta_i)\}_{i=1}^k$ policy is designed systematically.

**Theorem 3.1.** *Let* **s** *be the solution obtained by BSG (BSG-ABA or BSG-NE) for (2) with a given PAC parameters $(\nu, \delta)$ and stochastic greedy parameter $\epsilon \in (0, 1 - 1/e)$. Then, **s** is a $(\nu, \delta)$-PAC solution of (2) if the per-iteration $\{(\nu_i, \delta_i)\}_{i=1}^k$ policy belongs to the set*

$$\mathcal{A}(\nu, \delta) := \{\{(\nu_i, \delta_i)\}_{i=1}^k : \sum_{i=1}^k \delta_i = \delta, (1 - \epsilon) \sum_{i=1}^k \beta^{k-i} \nu_i \le \nu\}, \tag{6}$$

*where $\beta = 1 - (1 - \epsilon)/k$.*

*Proof.* The proof is provided in Section B.1. □

*Remark* 3.2. The ABA algorithm (Hassidim et al., 2020) requires its confidence parameter to lie in the range $(0, 0.05]$. Hence, we have the following constraint on user defined $\delta$ for BSG-ABA: $\delta \in \begin{cases} (0, 0.05k] & k < 20 \\ (0, 1) & k \ge 20 \end{cases}$.

Theorem 3.1 provides various options for designing the per-iteration $(\nu_i, \delta_i)$ policies for the SSM problem (2). For example, if we model $\nu_i = \nu_0, \delta_i = \frac{\delta}{k}, \forall i$ and solve for $\nu_0$ in $(1 - \epsilon) \sum_{i=1}^k \beta^{k-i} \nu_0 = \nu$, we obtain $(\nu_i, \delta_i) = (\frac{\nu}{k(1-\beta^k)}, \frac{\delta}{k})$ $\forall i$. Alternatively, since $\beta < 1$, one can upper bound $(1 - \epsilon) \sum_{i=1}^k \beta^{k-i} \nu_i$ with $(1 - \epsilon) \sum_{i=1}^k \nu_i$. If we equate the latter with $\nu$ and solve for $\nu_0$ (with $\nu_i = \nu_0$), we get another feasible policy $(\nu_i, \delta_i) = (\frac{\nu}{k(1-\epsilon)}, \frac{\delta}{k})$ $\forall i$. While Theorem 3.1 guarantees that both these example policies would allow BSG to obtain a $(\nu, \delta)$-PAC solution, the computational cost of BSG in terms of number of stochastic function $f$ calls $(N)$ would be different. Between the above two example policies, the former will incur lesser cost. In the following sections, we investigate learning per-iteration policies while taking the computational cost of BSG into consideration.

### 3.3 Computational cost of BSG

Theorem 3.1 characterizes a set of $\{(\nu_i, \delta_i)\}_{i=1}^k$ policies, $\mathcal{A}(\nu, \delta)$, which ensure that BSG obtains a $(\nu, \delta)$-PAC solution for the SSM problem (2). However, the computational cost, i.e., the number of stochastic function $f$ calls $(N)$, for obtaining a $(\nu, \delta)$-PAC solution depends on the policy used in BSG. For a given policy, the following lemma provides an expression of $N$ for the proposed BSG algorithms.

**Lemma 3.3.** *The number of stochastic function $f$ evaluations $N_{\mathrm{ABA}}$ and $N_{\mathrm{NE}}$ required by BSG-ABA and BSG-NE, respectively, are*

$$\begin{aligned} N_{\mathrm{ABA}} &= \sum_{i=1}^k \frac{18n}{k\nu_i^2} \log\left(\frac{1}{\delta_i}\right) \log\left(\frac{1}{\epsilon}\right), \\ N_{\mathrm{NE}} &= \sum_{i=1}^k \frac{2n}{k\nu_i^2} \log\left(\frac{n}{k\delta_i} \log\left(\frac{1}{\epsilon}\right)\right) \log\left(\frac{1}{\epsilon}\right). \end{aligned} \tag{7}$$

*Proof.* The proof is provided in Section B.2. □

*Remark* 3.4. The proof of computational complexity of BSG-ABA employs the result (Hassidim et al., 2020, Theorem 1) which provides an upper bound on the sample complexity of the ABA algorithm. In this work, we take this upper bound as the overall complexity of ABA.

Given $(\nu, \delta)$-PAC parameters for BSG, we now consider the problem of learning a policy from the set $\mathcal{A}(\nu, \delta)$ by minimizing the computational cost (7). The resulting optimization problem may be simplified as follows:

$$\min_{\nu_i, \delta_i > 0 \forall i} \quad \sum_{i=1}^k (\mu - \log(\delta_i)) \nu_i^{-2} \tag{8}$$
$$\text{s.t.} \quad \{(\nu_i, \delta_i)\}_{i=1}^k \in \mathcal{A}(\nu, \delta),$$

where $\mu = 0$ for BSG-ABA and $\mu = \log(\frac{n}{k} \log(\frac{1}{\epsilon}))$ for BSG-NE. Since $0 < \delta_i < 1$, $(\mu - \log(\delta_i)) > 0 \; \forall i$. While Problem (8) is non-convex, it is individually strictly convex in $\{\nu_i\}_{i=1}^k$ and $\{\delta_i\}_{i=1}^k$ variables and has closed form solutions. This is discussed in Appendix B.3. Hence, alternating minimization algorithm may be employed to solve (8).

In the following, we analyze (8) under the uniform per-iteration confidence setting, i.e., fixing $\delta_i = \delta/k \; \forall i$. This is motivated from the observation that the error parameters $\{\nu_i\}_{i=1}^k$ influence the computational cost (7) more significantly than the confidence parameters $\{\delta_i\}_{i=1}^k$. Having uniform per-iteration confidence also leads to simplified expressions.

**Theorem 3.5.** *Given $(\nu, \delta)$-PAC parameters, consider the uniform per-iteration confidence policies in $\mathcal{A}(\nu, \delta)$, i.e., $\{(\nu_i, \delta_i)\}_{i=1}^k \in \mathcal{A}(\nu, \delta)$ such that $\delta_i = \delta/k$. Among such policies, the one corresponding to the minimum number of stochastic function $f$ calls required by BSG to obtain a $(\nu, \delta)$-PAC solution is given by*

$$\nu_i^* = \frac{\nu(1 - \beta^{2/3})\beta^{(i-k)/3}}{(1 - \epsilon)(1 - \beta^{2k/3})} \; \forall i.$$

*The expressions of $N_{\mathrm{ABA}}$ and $N_{\mathrm{NE}}$ corresponding to $\{(\nu_i^*, \delta/k)\}_{i=1}^k$ are:*

$$N_{\mathrm{ABA}}^* = \frac{18n(1 - \epsilon)^2}{\nu^2} \log\left(\frac{k}{\delta}\right) \log\left(\frac{1}{\epsilon}\right) c(k),$$

$$N_{\mathrm{NE}}^* = \frac{2n(1 - \epsilon)^2}{\nu^2} \log\left(\frac{n}{\delta} \log\left(\frac{1}{\epsilon}\right)\right) \log\left(\frac{1}{\epsilon}\right) c(k),$$

*where $c(k) = \frac{(1 - \beta^{2k/3})^3}{k(1 - \beta^{2/3})^3}$ and $\beta = 1 - (1 - \epsilon)/k$.*

*Proof.* The proof involves solving (8) only with respect to $\{\nu_i\}_{i=1}^k$ by first setting $\delta_i = \delta/k$ and then using the Hölder's inequality. The details are provided in Section B.4. $\square$

*Remark* 3.6 (**Complexity of BSG-ABA and BSG-NE algorithms**). It can be shown that $c(k) \leq k^2$, resulting in linear $O(n)$ complexity of BSG-ABA and almost-linear $O(n \log(n))$ complexity of BSG-NE algorithms (ignoring the constants $\nu, \delta$, and $\epsilon$ and assuming $k \ll n$). This result is significant as for popular SSM problems such as exemplar-based clustering, both Greedy and Stochastic Greedy have quadratic $O(n^2)$ complexity in $k \ll n$ setting. From Table 1, we also observe that continuous stochastic greedy approaches (SGA, SCG, and SCG++) have at least quadratic $O(n^2)$ complexity.

*Remark* 3.7. Compared to Theorem 3.5, the margin of improvement with jointly optimizing both $\{\nu_i\}_{i=1}^k$ and $\{\delta_i\}_{i=1}^k$ in (8) is observed to be empirically small. The details are provided in Appendix D.

*Remark* 3.8. Recently, Tiwari et al. (2020) have proposed the BanditPAM algorithm for exemplar-based clustering ($k$-medoids) problem. BanditPAM employs a variant of UCB algorithm (Lai & Robbins, 1985; Zhang et al., 2019) and provides guarantees under probably correct setting (i.e., error parameter $\nu$ is 0). Given the confidence parameter $\delta$, BanditPAM has the following guarantee:
(Tiwari et al., 2020, Appendix Remark A1) With probability at least $1 - \delta$, BanditPAM, when restricted to the BUILD step returns the same set of $k$ points as Greedy, and the required number of pairwise distance computations satisfies $\mathbb{E}[N_{\mathrm{BP}}] = O(n^2 \delta d_1(k) + n \log(n/\delta) d_2(k))$.
Tiwari et al. (2020) derive the above result on the computational cost of BanditPAM ($N_{\mathrm{BP}}$) by assuming $k$ to be a constant and does not formally analyze its dependency on $k$. Hence, we have represented this dependency on $k$ as (unknown) functions $d_1(k)$ and $d_2(k)$ in the above expression. Incidentally, Tiwari et al. (2020, Appendix Section 2.5) observe that BanditPAM's computational cost varies from quadratic to cubic on $k$ for some parameter regime. Empirically, we observe that the computational cost of both BG and BSG algorithms are an order of magnitude lower than BanditPAM.

## 3.4  BSG in budget setting

We now study BSG in a budget setting. Given a budget of the number of stochastic function $f$ calls ($N_0$) and the overall confidence parameter $\delta$, the aim is to find the best per-iteration policy in the set $\mathcal{A}(\nu, \delta)$. Since

the overall PAC-error $\nu$ parameter for BSG is not provided in this setting and the best policy for BSG is the one which achieves the lowest $\nu$ for the SSM problem (2).

The above may be posed as an optimization problem over the policy $\{(\nu_i, \delta_i)\}_{i=1}^k \in \mathcal{A}(\nu, \delta)$ such that $\nu$ is minimized. We begin by noting that given a policy $\{(\nu_i, \delta_i)\}_{i=1}^k \in \mathcal{A}(\nu, \delta)$, the best (lowest) PAC-error parameter (for BSG) is a function of the policy: $\nu = (1 - \epsilon) \sum_{i=1}^k \beta^{k-i} \nu_i$. Hence, we propose to minimize $\nu$ over the set of policies $\mathcal{A}(\nu, \delta)$ and under the constraints that the overall budget is $N_0$ and the overall PAC-confidence parameter for BSG is $\delta$. The resulting optimization problem may be simplified as follows:

$$
\min_{\nu_i, \delta_i > 0 \forall i} \quad (1 - \epsilon) \sum_{i=1}^k \beta^{k-i} \nu_i
$$
$$
\text{s.t.} \quad \sum_{i=1}^k (\mu - \log(\delta_i)) \nu_i^{-2} \leq c, \quad \sum_{i=1}^k \delta_i \leq \delta, \tag{9}
$$

where $\mu = 0, c = \frac{kN_0}{18n \log(1/\epsilon)}$ for BSG-ABA and $\mu = \log(\frac{n}{k} \log(\frac{1}{\epsilon})), c = \frac{kN_0}{2n \log(1/\epsilon)}$ for BSG-NE. Since $0 < \delta_i < 1$, $(\mu - \log(\delta_i)) > 0 \; \forall i$. Problem (9) is a non-convex and we may employ a gradient descent based algorithm. This is discussed in Appendix B.6.

In the following, we analyze (9) under the uniform per-iteration confidence setting, i.e., fixing $\delta_i = \delta/k \; \forall i$. This leads to simple closed-form expression for the optimal policy and has only a marginal empirical difference compared to directly optimizing (9). This is discussed in detail in Appendix D.

**Theorem 3.9.** *Given a budget $N_0$ on the number of stochastic function $f$ calls, the confidence parameter $\delta$ for BSG, and under the uniform per-iteration confidence setting (i.e., $\delta_i = \delta/k \; \forall i$), BSG-ABA and BSG-NE obtain $(\nu_{\text{ABA}}^*, \delta)$- and $(\nu_{\text{NE}}^*, \delta)$-PAC solutions, respectively, where*

$$
\nu_{\text{ABA}}^* = \left( \frac{1 - \beta^{2k/3}}{1 - \beta^{2/3}} \right)^{3/2} \left( \frac{(1 - \epsilon)^2 18n \log(k/\delta) \log(1/\epsilon)}{kN_0} \right)^{1/2},
$$
$$
\nu_{\text{NE}}^* = \left( \frac{1 - \beta^{2k/3}}{1 - \beta^{2/3}} \right)^{3/2} \left( \frac{(1 - \epsilon)^2 2n \log(n \log(1/\epsilon)/\delta) \log(1/\epsilon)}{kN_0} \right)^{1/2},
$$

*and $\beta = 1 - (1 - \epsilon)/k$. The corresponding per-iteration error levels for BSG-ABA is*

$$
\nu_i^* = \left( \frac{18n \log(k/\delta) \log(1/\epsilon)(1 - \beta^{2k/3})}{kN_0(1 - \beta^{2/3})} \right)^{1/2} \beta^{(i-k)/3} \; \forall i.
$$

*Similarly, corresponding per-iteration error levels for BSG-NE is*

$$
\nu_i^* = \left( \frac{2n \log(n \log(1/\epsilon)/\delta) \log(1/\epsilon)(1 - \beta^{2k/3})}{kN_0(1 - \beta^{2/3})} \right)^{1/2} \beta^{(i-k)/3} \; \forall i.
$$

*Proof.* The proof involves solving (9) only with respect to $\{\nu_i\}_{i=1}^k$ by first setting $\delta_i = \delta/k$ and then using the Hölder's inequality. The details are provided in Appendix B.7. □

### 3.5 Implementation details

We now discuss a few implementation-related aspects of the proposed BSG approach.

**Computing rewards for all the arms on the same sample set.** In a given iteration $i$ of BSG (Algorithm 1), let $\mathcal{Z}_i$ be the set of instances $\mathbf{z}$ sampled for the first arm. Then, the same set $\mathcal{Z}_i$ may be used for computing rewards for the remaining candidate arms. This offers several practical advantages without changing any theoretical results: (a) reduces the required number of samples in an iteration (by a factor of the number of arms), which becomes important when acquiring samples is costly, and (b) helps in memory-efficient parallelized computations. However, we note that using the same set $\mathcal{Z}_i$ *does not* reduce the number of calls to the stochastic function $f$, as rewards needs to be computed for all the candidate arms.

**On BSG-ABA versus BSG-NE.** In Remark 3.6, we observe that BSG-ABA has lower complexity of $O(n)$ on required number of stochastic function $f$ calls than BSG-NE which has complexity $O(n \log(n))$ for a given $(\nu, \delta)$-PAC setting. However, from (7), we note that the constant terms hiding in the $O$ notation of BSG-ABA are larger than those of BSG-NE ($N^*_{\text{ABA}}$ has 18 while $N^*_{\text{NE}}$ has 2). By setting $N^*_{\text{NE}} \leq N^*_{\text{ABA}}$, we can therefore compute the range $[1, n_0]$ of $n$ over which BSG-NE requires lower number of stochastic function $f$ calls than BSG-ABA. We get $n_0 = \delta(k/\delta)^9 / \log(1/\epsilon)$. As discussed in Remark 3.2, $\delta/k \leq 0.05$ for BSG-ABA. Hence, $n_0 \geq 5.12 \times 10^{12} \frac{\delta}{\log(1/\epsilon)}$. For reasonable choice of parameter values $\delta = 0.001$ and $\epsilon = 0.01$, we get $n_0 \geq 2.3 \times 10^9$. Hence, BSG-ABA will have a practical runtime advantage over BSG-NE only in very large scale problem setting.

Overall, we observe that BSG-NE is a well-suited algorithm for the SSM problem (2) as it has lower stochastic function calls complexity than existing SSM algorithms like SGA (Hassani et al., 2017), SCG (Mokhtari et al., 2018), and SCG++ (Hassani et al., 2019). BSG-NE is simple to implement, can take advantage of lazy evaluations (Minoux, 1978), and can parallelize the reward computation across the candidate arms and samples within each iteration. This allows BSG-NE to exploit modern GPU computing systems, thereby making BSG-NE (and for the same reason BG-NE) our method of choice for experiments discussed in Section 4.

## 4 Experiments

In this section, we show the benefit of the proposed BSG and BG algorithms on the exemplar-based clustering and representative sampling applications. As discussed, we consider the BSG-NE and BG-NE variants of the proposed Algorithm 1. For both our algorithms, we pick the $\{(\nu_i, \delta_i)\}_{i=1}^k$ policy (Theorem 3.5) that minimize the stochastic function $f$ calls for the given $(\nu, \delta)$ parameters. We consider the following baselines.

1. **LG**: the deterministic Greedy algorithm with lazy evaluations (Minoux, 1978; Leskovec et al., 2007). As it has the best theoretical guarantee, LG is expected to obtain the best generalization performance among the competing methods but with high computational cost.

2. **LSG**: the popular Stochastic Greedy algorithm with lazy evaluations (Mirzasoleiman et al., 2015).

3. **SCG:** the stochastic continuous gradient algorithm proposed in (Mokhtari et al., 2018; 2020).

4. **BanditPAM:** UCB best-arm identification based $k$-medoids method (Tiwari et al., 2020). We run only the BUILD step of BanditPAM, which corresponds to applying SSM to the exemplar-based clustering problem. The authors' C++ code link: `https://github.com/motiwari/BanditPAM`.

All the algorithms (except BanditPAM) are implemented in Matlab. The experiments are run on Intel Xeon CPU (3.6 GHz) with 6 cores and 64 GB RAM. The following datasets are considered.

- **MNIST** (LeCun et al., 1998) is a handwritten digits dataset with $28 \times 28$ pixels greyscale images of 10 classes for digits $\{0, 1, \ldots 9\}$. It has two different sets of $60,000$ samples (train) and $10,000$ samples (test).

- **TinyImageNet (TIN)** is a smaller version of the ImageNet dataset (Russakovsky et al., 2015) with 200 classes and 500 instances per class (Wu et al., 2017). It consists of images of size $64 \times 64$ pixels.

**Exemplar-based clustering.** Given a dataset $\mathcal{X} = \{\mathbf{x}_i\}_{i=1}^n$, the aim in exemplar-based clustering is to select $k \ll n$ relevant data points $\mathbf{s} \subseteq \mathcal{X}$ that best represent the dataset $\mathcal{X}$. A popular approach is the $k$-medoids formulation (Kaufman & Rousseeuw, 2009) that aims to minimize the average distance of the data points to their nearest exemplars, i.e., minimize $L(\mathbf{s}) := \frac{1}{n} \sum_{\mathbf{z} \in \mathcal{X}} \min_{\mathbf{x} \in \mathbf{s}} \text{dist}(\mathbf{x}, \mathbf{z})$. Here, $\text{dist}(\mathbf{x}, \mathbf{z})$ computes the distance between the data points $\mathbf{x}$ and $\mathbf{z}$. We can pose the above minimization problem as an equivalent monotone (stochastic) submodular maximization problem by introducing a phantom exemplar $\mathbf{z}_0$ (Gomes & Krause, 2010): $F(\mathbf{s}) := L(\mathbf{z}_0) - L(\mathbf{s} \cup \{\mathbf{z}_0\})$. Thus, maximizing $F(\cdot)$ is equivalent to minimizing $L(\cdot)$. We can

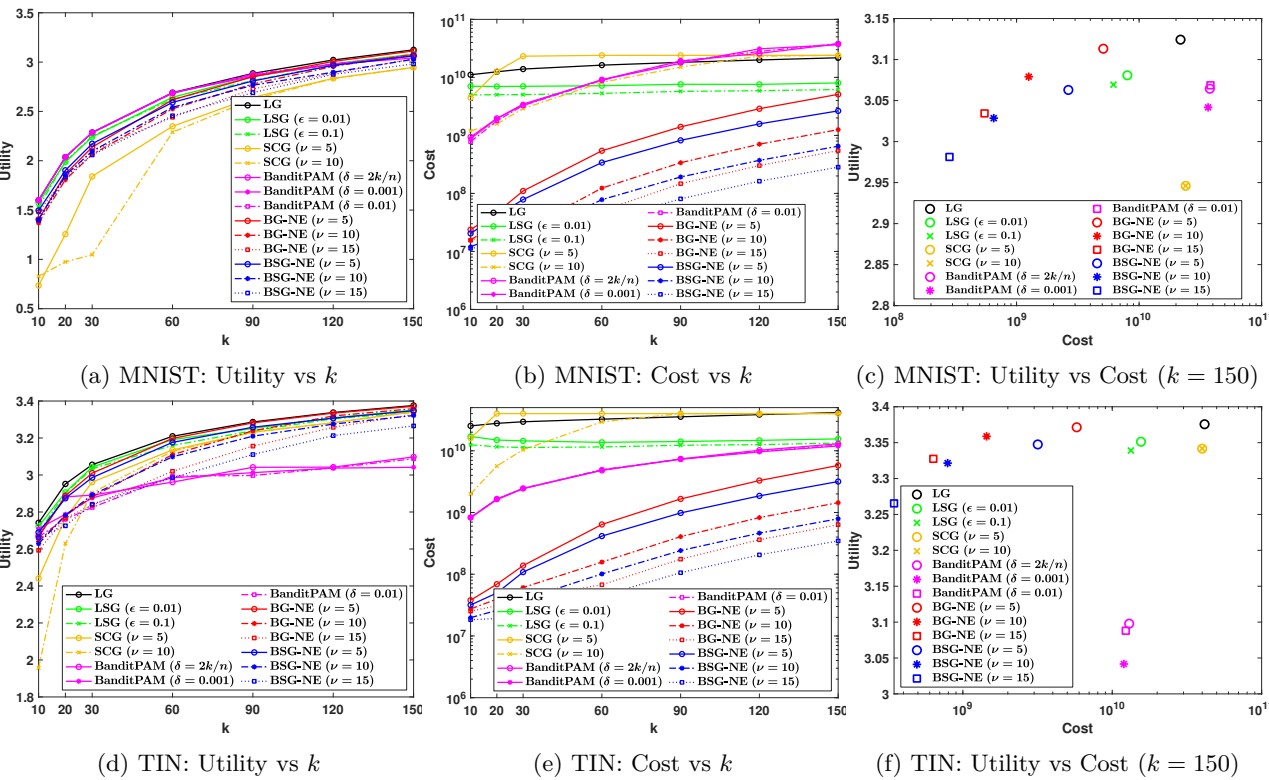

(a) MNIST: Utility vs $k$      (b) MNIST: Cost vs $k$      (c) MNIST: Utility vs Cost ($k = 150$)

(d) TIN: Utility vs $k$      (e) TIN: Cost vs $k$      (f) TIN: Utility vs Cost ($k = 150$)

Figure 1: Performance of various algorithms on the exemplar-based clustering problem. Top row: MNIST dataset, Bottom row: TIN dataset. We observe that the proposed algorithms BSG and BG obtain a good utility versus cost trade-off by varying error parameter $\nu$ and outperforms other SSM methods such as LSG, SCG, and BanditPAM.

select the phantom exemplar as any point $\mathbf{z}_0$ satisfying the condition $\max_{\mathbf{z}' \in \mathcal{X}} \mathrm{dist}(\mathbf{z}', \mathbf{z}) \le \mathrm{dist}(\mathbf{z}_0, \mathbf{z}) \ \forall \mathbf{z} \in \mathcal{X}$ (Mirzasoleiman et al., 2016). This ensures that $L(\mathbf{s} \cup \{\mathbf{z}_0\}) = L(\mathbf{s})$. The stochastic function $f$ can then be expressed as $f(\mathbf{s}; \mathbf{z}) = L(\mathbf{z}_0) - \min_{\mathbf{x} \in \mathbf{s}} \mathrm{dist}(\mathbf{x}, \mathbf{z})$. In our experiments, we set $\mathrm{dist}(\mathbf{x}, \mathbf{z}) = \|\mathbf{x} - \mathbf{z}\|^2$.

For BSG-NE, BG-NE, and SCG, we experiment with $\nu = \{5, 10, 15\}$. As the error parameter $\nu$ increases, these methods require lesser number of samples (BSG-NE and BG-NE) or iterations (SCG). The number of iterations for SCG is upper bounded by $20\,000$. We set $\delta = 0.001$ for BSG-NE and BG-NE and $\epsilon = 0.01$ for BSG-NE. We experiment with LSG in two settings: $\epsilon = 0.01$ and $\epsilon = 0.1$. For BanditPAM, we show results with $\delta = \{0.01, 0.001, 2k/n\}$. The per iteration policy of BanditPAM is $\delta_i = \delta/k$ (Tiwari et al., 2020).

Figure 1 top-row and Figure 1 bottom-row show the results on the MNIST and TIN datasets, respectively. From Figures 1(a) & 1(d), we observe that the proposed BSG-NE and BG-NE algorithms' utility is comparable with LG across $k$, especially at lower values of error parameter $\nu$. BSG-NE and BG-NE methods achieve the lowest cost across $k$ in both the datasets, i.e., in Figures 1(b) & 1(e), where cost is the number of stochastic function $f$ calls. In exemplar-based clustering, this cost is the number of pairwise distance computations done by an algorithm. At a high $k$ value of 150, we observe in Figures 1(c) & 1(f) that BG-NE with $\nu = 5$ obtains the closest utility to LG and is computationally cheaper than both LG and LSG. On both datasets, across $k$, the proposed BSG-NE and BG-NE methods show a good trade-off between utility and cost with varying error parameter $\nu$. LSG's cost, on the other hand, varies little across $k$ and is much costlier than the proposed approaches at low $k$ values. SCG obtains the worst utility across $k$ on the MNIST dataset and is lower than the proposed approaches on the TIN dataset. BanditPAM, on the other hand, matches LG's utility at low $k$ values on MNIST, but as $k$ increases, it ends up with utility lower than both BG and BSG algorithms. On the TIN dataset, BanditPAM obtains a competitive utility to BG/BSG (with $\nu = 15$) till $k = 20$ but its utility almost flattens for $k > 30$. The computational cost of SCG and BanditPAM are also

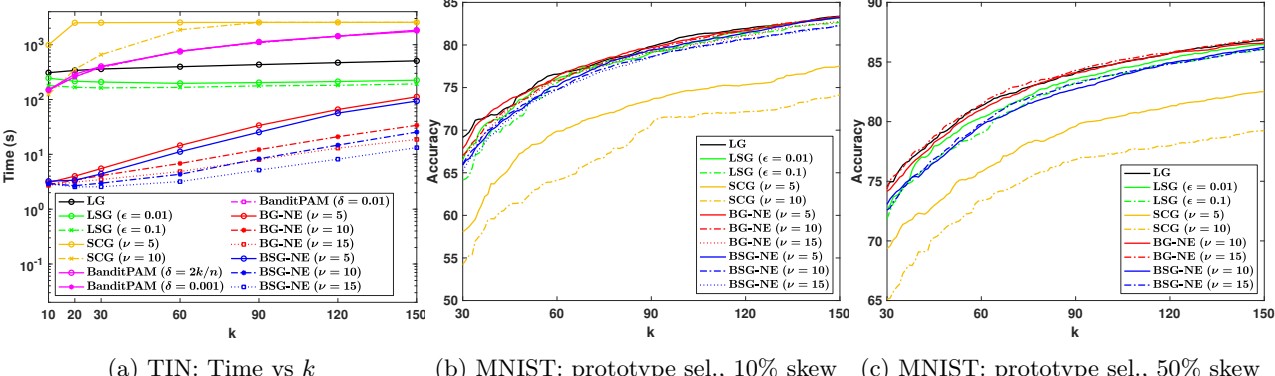

(a) TIN: Time vs $k$      (b) MNIST: prototype sel., 10% skew    (c) MNIST: prototype sel., 50% skew

Figure 2: (a) Time versus $k$ plot for the exemplar-based clustering problem on the TIN dataset; (b)&(c) Generalization performance achieved by nearest prototype classifier corresponding to each method on representative sampling problem.

significantly higher than BSG-NE and BG-NE. In Figures 1(c) & 1(f), we observe that SCG and BanditPAM obtain the worst utility versus cost trade-off on the MNIST and TIN datasets, respectively. In Figure 2(a), we also plot the time taken for different $k$ values on TIN. We observe that BSG-NE and BG-NE take much less time than the baselines, highlighting their practical usefulness.

**Representative sampling from a target set.** The previous set of experiments show that the proposed algorithms provide a better utility versus cost trade-off compared to the existing approaches. In the above experiments, the quality of the obtained solution is evaluated only in terms of the achieved (submodular) objective. We now evaluate the quality of the proposed approaches in terms of the generalization performance of the obtained solution in a downstream application).

The aim in this application is to select few $k \ll n$ data points from a given *source* set $\mathcal{X}$ that best represents a different given *target* set $\mathcal{Z}$ (Kim et al., 2016). It can also be viewed as an instance of the facility location problem and is formulated as maximizing $\sum_{\mathbf{z} \in \mathcal{Z}} (\max_{\mathbf{x} \in \mathbf{s}_\mathcal{X}} w(\mathbf{x}, \mathbf{z}))$, where $w$ computes the similarity between points from $\mathcal{X}$ and $\mathcal{Z}$. Hence, the above representative sampling formulation is an instance of SSM problem. Following (Bien & Tibshirani, 2011; Kim et al., 2016), we evaluate the quality of the selected representative samples (a.k.a. prototypes) via the performance of the corresponding nearest prototype classifier. This is a 1-NN classifier which is parameterized by the selected prototypes.

We evaluate all the methods on the MNIST dataset using the standard experimental protocol (Gurumoorthy et al., 2019; 2021). The source set consists of $5,000$ points uniformly sampled from the MNIST test set (of size $10,000$). The target set is constructed from the MNIST train set consisting of $60,000$ points such that one class has a skewed $r\%$ representation and others have $(100 - r)/9\%$ representation each. We evaluate the methods in two skew settings: $r = 10$ and $r = 50$. We compare BSG-NE and BG-NE against the baselines LG, LSG, and SCG. BanditPAM is not evaluated as its code is made available only for the $k$-medoids problem.

The accuracy of the nearest prototype classifier corresponding to each method is plotted in Figures 2(b) & 2(c). We observe that the prototypes selected by our BSG-NE and BG-NE algorithms show good generalization performance and are comparable to LG. This shows that the quality of the solution obtained by the proposed algorithms is similar to LG's solution. However, the prototypes selected by SCG are less informative as they obtain $8\% - 15\%$ less accuracy than our approach.

## 5   Conclusion

In this paper we took a fresh look at the stochastic submodular maximization (SSM) problem from a bandit viewpoint. Each iteration of the greedy framework is naturally posed as a best-arm selection (pure exploration) problem. Subsequently, we make use of the best-arm identification strategy at each iteration $i$ to select a

$(\nu_i, \delta_i)$-PAC best element, where $\nu_i$ is the error parameter and $\delta_i$ is the confidence parameter. Theoretically, we formalize a set of $\{(\nu_i, \delta_i)\}_{i=1}^k$ policies which ensure that our proposed algorithms obtain a $(\nu, \delta)$-PAC solution. We next investigate learning policies for: (i) minimizing the computational cost, and (ii) ensuring that the computational cost is within a given budget. With the learned policies, our algorithms have either linear or almost-linear computational cost in the size of the problem $n$. Overall, our algorithms offer better utility versus computational cost trade-off than existing SSM approaches in both theory and practice.

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

# Appendix

## A    Organization

The appendix is organized as follows.

1. Section B gives the proofs to the theorems and lemmas mentioned in the main paper.

2. Section C discusses the theorems and the proofs related to the correctness of the proposed BG algorithm.

3. Additional discussion on the algorithms is presented in Section E.

4. The experimental details are in Section F.

5. The bandit algorithms for PAC_BEST_ARM in Algorithm 1 are in Section G.

## B    Proofs related to the BSG algorithm

### B.1    Proof of Theorem 3.1

We first note the following lemma which adapts (Mirzasoleiman et al., 2015, Lemma 2) to the $(\nu, \delta)$-PAC best arm identification setting and then use this result to prove Theorem 3.1.

**Lemma B.1.** *Given a current solution* $\mathbf{s}$ *of BSG, let* $(\nu_0, \delta_i)$ *be the PAC parameters for the next iteration. Then, the following holds with probability* $1 - \delta_i$*: the expected gain of BSG (Algorithm 1) in one iteration is at least* $\frac{(1-\epsilon)}{k} \sum_{s \in \mathbf{s}^* \setminus \mathbf{s}} (g(s|\mathbf{s}) - \nu_0)$ *where* $\mathbf{s}^*$ *denote the optimal solution of problem (2).*

*Proof.* Let $\mathcal{R}$ denote the set of candidate elements in the current iteration. We first estimate the probability that $\mathcal{R} \cap (\mathbf{s}^* \setminus \mathbf{s})$ is non-empty. The set $\mathcal{R}$ has $r = \frac{n}{k} \log(\frac{1}{\epsilon})$ randomly sampled elements from set $\mathcal{S} \setminus \mathbf{s}$ (w.l.o.g. with repetition). Hence,

$$
\begin{aligned}
\mathbb{P}[\mathcal{R} \cap (\mathbf{s}^* \setminus \mathbf{s}) \neq \phi] \quad &= 1 - \mathbb{P}[\mathcal{R} \cap (\mathbf{s}^* \setminus \mathbf{s}) = \phi] \\
&= 1 - \left(1 - \frac{|\mathbf{s}^* \setminus \mathbf{s}|}{|\mathcal{S} \setminus \mathbf{s}|}\right)^r \\
&\geq 1 - e^{-r \frac{|\mathbf{s}^* \setminus \mathbf{s}|}{|\mathcal{S} \setminus \mathbf{s}|}} & (\because 1 - x \leq e^{-x}) \\
&\geq 1 - e^{-\frac{r}{n}|\mathbf{s}^* \setminus \mathbf{s}|} & (\because |\mathcal{S} \setminus \mathbf{s}| \leq n).
\end{aligned}
$$

Since $1 - e^x$ is a concave function and $|\mathbf{s}^* \setminus \mathbf{s}|/k \leq 1$, we have

$$
1 - e^{-\frac{r}{n}\left((1 - \frac{|\mathbf{s}^* \setminus \mathbf{s}|}{k})0 + (\frac{|\mathbf{s}^* \setminus \mathbf{s}|}{k})k\right)} \geq ((1 - \frac{|\mathbf{s}^* \setminus \mathbf{s}|}{k}))(1 - e^{-\frac{r}{n}0}) + (\frac{|\mathbf{s}^* \setminus \mathbf{s}|}{k})(1 - e^{-\frac{r}{n}k}),
$$

which reduces to

$$
1 - e^{-\frac{r}{n}|\mathbf{s}^* \setminus \mathbf{s}|} \geq \frac{|\mathbf{s}^* \setminus \mathbf{s}|}{k}(1 - e^{-\frac{rk}{n}}).
$$

Therefore, we have

$$
\mathbb{P}[\mathcal{R} \cap (\mathbf{s}^* \setminus \mathbf{s}) \neq \phi] \geq 1 - e^{-\frac{r}{n}|\mathbf{s}^* \setminus \mathbf{s}|} \geq \frac{|\mathbf{s}^* \setminus \mathbf{s}|}{k}(1 - e^{-\frac{rk}{n}}) = (1 - \epsilon)\frac{|\mathbf{s}^* \setminus \mathbf{s}|}{k}. \tag{10}
$$

In a given iteration, BSG's step 2 (`PAC_BEST_ARM`) selects a $\nu_0$-optimal element $s$ with probability $1 - \delta_i$. That is,

$$g(s|\mathbf{s}) \geq \max_a g(a|\mathbf{s}) - \nu_0. \tag{11}$$

with probability at least $1 - \delta_i$. In the following, we omit repeating the phrase "with probability at least $1 - \delta_i$" with every result/implication for brevity. Equation (11) implies

$$g(s|\mathbf{s}) \geq g(s|\mathbf{s}) - \nu_0 \ \forall s \in \mathcal{R} \cap (\mathbf{s}^* \setminus \mathbf{s}) \text{ (if non-empty) with probability } 1 - \delta_i.$$

We note that since $\mathcal{R}$ is equally likely to contain any element of $\mathbf{s}^* \setminus \mathbf{s}$, a uniformly random element of $\mathcal{R} \cap (\mathbf{s}^* \setminus \mathbf{s})$ is a uniformly random element of $\mathbf{s}^* \setminus \mathbf{s}$. Taking expectation on both sides of the inequality in (11) and using the above observation, we get

$$\mathbb{E}[g(s|\mathbf{s})] \geq \mathbb{P}[\mathcal{R} \cap (\mathbf{s}^* \setminus \mathbf{s}) \neq \phi] \left( \frac{1}{|\mathbf{s}^* \setminus \mathbf{s}|} \sum_{s \in \mathbf{s}^* \setminus \mathbf{s}} (g(s|\mathbf{s}) - \nu_0) \right)$$

The above inequality holds as the expectation is being taken over the events $\mathbb{P}[\mathcal{R} \cap (\mathbf{s}^* \setminus \mathbf{s}) \neq \phi]$ and $\mathbb{P}[\mathcal{R} \cap (\mathbf{s}^* \setminus \mathbf{s}) = \phi]$ and the product of marginal gain and the latter event is non-negative. By substituting (10) in the right hand side of the above inequality, we get that the following holds with probability at least $1 - \delta_i$

$$\mathbb{E}[g(s|\mathbf{s})] \geq \frac{1 - \epsilon}{k} \sum_{s \in \mathbf{s}^* \setminus \mathbf{s}} (g(s|\mathbf{s}) - \nu_0).$$

This completes the proof of Lemma B.1. $\qquad\square$

**Now, the proof of Theorem 3.1 begins.**

Let $\mathbf{s}_i = \{s_1, \ldots, s_i\}$ be the solution of BSG after $i$ iterations. Let $(\nu_i, \delta_i)$ be a feasible per-iteration policy for BSG such that employing it ensures that BSG learns a $(\nu, \delta)$-PAC solution. From Lemma B.1, we have the following result:

$$\mathbb{E}[g(s_{i+1}|\mathbf{s}_i)|\mathbf{s}_i] \geq \frac{1 - \epsilon}{k} \sum_{s \in \mathbf{s}^* \setminus \mathbf{s}_i} (g(s|\mathbf{s}_i) - \nu_{i+1}), \tag{12}$$

which holds with probability at least $1 - \delta_i$. By submodularity, we have the following result: $\sum_{s \in \mathbf{s}^* \setminus \mathbf{s}} g(s|\mathbf{s}_i) \geq g(\mathbf{s}^*|\mathbf{s}_i) \geq F(\mathbf{s}^*) - F(\mathbf{s}_i)$. Applying this in the above result gives

$$\mathbb{E}[g(s_{i+1}|\mathbf{s}_i)|\mathbf{s}_i] \geq \frac{1 - \epsilon}{k} (F(\mathbf{s}^*) - F(\mathbf{s}_i)) - (1 - \epsilon)\nu_{i+1}.$$

Now we take expectation over $\mathbf{s}_i$ over both the sides and using the law of total expectation, we have

$$\mathbb{E}[g(s_{i+1}|\mathbf{s}_i)] = \mathbb{E}[F(\mathbf{s}_{i+1}) - F(\mathbf{s}_i)] \geq \frac{1 - \epsilon}{k} \mathbb{E}[F(\mathbf{s}^*) - F(\mathbf{s}_i)] - (1 - \epsilon)\nu_{i+1}.$$

Since $\mathbb{E}[F(\mathbf{s}^*)] = F(\mathbf{s}^*)$, we get

$$(F(\mathbf{s}^*) - \mathbb{E}[F(\mathbf{s}_i)]) - (F(\mathbf{s}^*) - \mathbb{E}[F(\mathbf{s}_{i+1})]) \geq \frac{1 - \epsilon}{k} (F(\mathbf{s}^*) - \mathbb{E}[F(\mathbf{s}_i)]) - (1 - \epsilon)\nu_{i+1}.$$

Taking $B_{i+1} = F(\mathbf{s}^*) - \mathbb{E}[F(\mathbf{s}_{i+1})]$ and $\beta = (1 - \frac{1-\epsilon}{k})$ we get

$$B_{i+1} \leq \beta B_i + (1 - \epsilon)\nu_{i+1}.$$

By induction, we have

$$B_k \leq \beta^k B_0 + (1 - \epsilon) \sum_{i=1}^{k} \beta^{k-i} \nu_i. \tag{13}$$

If $(1 - \epsilon) \sum_{i=1}^{k} \beta^{k-i} \nu_i \leq \nu$, the inequality (13) implies

$$\mathbb{E}[F(\mathbf{s}_k)] \geq (1 - e^{-1} - \epsilon) F(\mathbf{s}^*) - \nu, \tag{14}$$

where we also use the relation that $(1 - (1 - \frac{1-\epsilon}{k})^k) \geq (1 - e^{-(1-\epsilon)}) \geq (1 - e^{-1} - \epsilon)$.

Thus, the result (14) is true when both the following conditions hold:

1. $(1 - \epsilon) \sum_{i=1}^{k} \beta^{k-i} \nu_i \leq \nu$

2. the result (12) is true for every iteration of BSG (Algorithm 1)

However, (12) depends on the validity of (11). Now, BSG's step 2 returns a $\nu_i$-optimal solution with probability $1 - \delta_i$. The probability that BSG's step 2 does not return $\nu_i$-optimal solution in any iteration $i \in \{1, \ldots, k\}$ is $\sum_i \delta_i = \delta$. Hence, the probability that BSG's step 2 return $\nu_i$-optimal solution in every iteration $i$ for $i = 1, \ldots, k$ is $1 - \delta$. Therefor, we prove that (14) is true with probability $1 - \delta$ if $(1 - \epsilon) \sum_{i=1}^{k} \beta^{k-i} \nu_i \leq \nu$ and $\sum_i \delta_i = \delta$, which completes the proof.

## B.2 Proof of Lemma 3.3

The $i$-th iteration of BSG-ABA calls the ABA algorithm once to obtain a $\nu_i, \delta_i$-PAC solution. For $n$ arms, the number of samples $\mathbf{z}$ required by ABA algorithm (Hassidim et al., 2020, Theorem 1) to select a $\nu_i$-best arm with probability at least $1 - \delta_i$ is at most $\frac{18n}{\nu^2} \log(\frac{1}{\delta})$. One stochastic function $f$ call is required for every sample $\mathbf{z}$. The candidate set $\mathcal{R}_i$ in each iteration $i$ of BSG-ABA has $|\mathcal{R}_i| = \frac{n}{k} \log(\frac{1}{\epsilon})$ elements. Hence, total number of function calls required by BSG-ABA across $k$ iterations is

$$N_{\text{ABA}} = \sum_{i=1}^{k} \frac{18|\mathcal{R}_i|}{\nu_i^2} \log\left(\frac{1}{\delta_i}\right) = \sum_{i=1}^{k} \frac{18n}{k\nu_i^2} \log\left(\frac{1}{\delta_i}\right) \log\left(\frac{1}{\epsilon}\right).$$

For $n$ arms, the number of samples $\mathbf{z}$ required by the NE algorithm to select a $\nu_i$-best arm with probability at least $1 - \delta_i$ is $\frac{2n}{\nu^2} \log(\frac{n}{\delta})$. This well-known result can be proved using the Hoeffding bound (Hassidim et al., 2020, Appendix A). The $i$-th iteration of BSG-NE calls the NE algorithm once to obtain a $\nu_i, \delta_i$-PAC solution. Hence, the total number of function calls required by BSG-NE across $k$ iterations is

$$N_{\text{NE}} = \sum_{i=1}^{k} \frac{2|\mathcal{R}_i|}{\nu_i^2} \log\left(\frac{|\mathcal{R}_i|}{\delta_i}\right) = \sum_{i=1}^{k} \frac{2n}{k\nu_i^2} \log\left(\frac{n}{k\delta_i} \log\left(\frac{1}{\epsilon}\right)\right) \log\left(\frac{1}{\epsilon}\right).$$

## B.3 Alternating minimization algorithm for (8)

We first state the following result (Micchelli & Pontil, 2005, Lemma 26).

**Lemma B.2.** *Let $r > 0$, $p = 1 + \frac{1}{r}$, $a_i > 0$ $\forall i = 1, \ldots, k$. Consider the below optimization problem:*

$$\min_{x_j > 0, j=1,\ldots,k, \sum_j x_j^r \leq 1} \sum_{i=1}^{k} \frac{a_i^2}{x_i}. \tag{15}$$

*The optimal objective of the above problem is $\left(\sum_{i=1}^{k} a_i^{\frac{2}{p}}\right)^p$ which occurs at*

$$x_i^* = \left(\frac{a_i^{\frac{2}{p}}}{\sum_j a_j^{\frac{2}{p}}}\right)^{\frac{1}{r}}. \tag{16}$$

*Proof.* Lemma B.2 can be proved using the Hölder inequality (Micchelli & Pontil, 2005, Lemma 26). □

### B.3.1 Solve (8) for given $\{\delta_i\}_{i=1}^k$

We first solve for $\{\nu_i\}_{i=1}^k$ in (8) for given $\{\delta_i\}_{i=1}^k$. The optimization problem (8) corresponding to this setting can be rewritten as

$$\min_{\nu_i > 0 \forall i} \quad \sum_{i=1}^k (\mu - \log(\delta_i))\nu_i^{-2}$$

$$\text{s.t.} \quad (1 - \epsilon)\sum_{i=1}^k \beta^{k-i}\nu_i \leq \nu. \tag{17}$$

To solve this, we first use the substitution

$$\sqrt{x_i} = (1 - \epsilon)\beta^{k-i}\nu_i/\nu.$$

This leads to the following equivalent optimization problem:

$$\min_{x_i > 0 \forall i} \quad \sum_{i=1}^k \left(\frac{(1-\epsilon)\sqrt{\mu - \log(\delta_i)}\beta^{k-i}}{\nu}\right)^2 \frac{1}{x_i}$$

$$\text{s.t.} \quad \sum_{i=1}^k \sqrt{x_i} \leq 1. \tag{18}$$

We now make use of Lemma B.2. The expression of optimal $\nu_i^*$ for given $\{\delta_i\}_{i=1}^k$ is

$$\nu_i^* = \frac{\nu\sqrt{x_i^*}}{(1-\epsilon)\beta^{k-i}} = \frac{\nu a_i^{\frac{2}{p}}}{(1-\epsilon)\beta^{k-i}\sum_j a_j^{\frac{2}{p}}}, \tag{19}$$

where $x_i^*$ is given by (16), $a_i = \frac{(1-\epsilon)\sqrt{\mu - \log(\delta_i)}\beta^{k-i}}{\nu}$, $r = 1/2$, and $p = 3$.

### B.3.2 Solve (8) for given $\{\nu_i\}_{i=1}^k$

We now solve for $\{\delta_i\}_{i=1}^k$ in (8) for given $\{\nu_i\}_{i=1}^k$. The optimization problem (8) corresponding to this setting can be rewritten as

$$\min_{\delta_i > 0 \forall i} \quad \sum_{i=1}^k -\log(\delta_i)\nu_i^{-2}$$

$$\text{s.t.} \quad \sum_{i=1}^k \delta_i \leq \delta. \tag{20}$$

This can be solved using Lagragian duality and at optimality, $\delta_i^* = \frac{\delta}{\nu_i^2}\left(\sum_j \frac{1}{\nu_j^2}\right)^{-1}$.

### B.4 Proof of Theorem 3.5

For this, we simply need to solve Problem (8) only with respect to $\{\nu_i\}_{i=1}^k$ and fix $\delta_i = \delta/k \ \forall i$. Appendix B.3.1 discuss this optimization problem. Hence, we get the expression of optimal $\nu_i^* \ \forall i$ by setting $\delta_i = \delta/k \ \forall i$ in (19). Plugging these values of $\{\nu_i^*\}_{i=1}^k$ and $\{\delta_i\}_{i=1}^k$ back in $N_{\text{ABA}}$ and $N_{\text{NE}}$ (7) completes the proof of Theorem 3.5.

### B.5 More details on $c(k) \leq k^2$ in Remark 3.6

The expression of $c(k)$ is $c(k) = \frac{(1-\beta^{2k/3})^3}{k(1-\beta^{2/3})^3}$. We consider the function

$$c_1(k) = (c(k)/k^2)^{1/3} = \frac{(1-\beta^{2k/3})}{k(1-\beta^{2/3})}.$$

It is easy to see that if $c_1(k) < 1$, it implies that $c(k) \leq k^2$. Hence, we aim to prove $c_1(k) < 1$ in the following way.

- The binomial theorem for fractional exponent is given by $(1+x)^{\frac{p}{q}} = 1 + \frac{p}{q}x + \frac{\frac{p}{q}(\frac{p}{q}-1)}{2!}x^2 + \ldots = \sum_{k=0}^{\infty} \binom{p/q}{k} x^k$. Hence,

$$k(1 - \frac{1-\epsilon}{k})^{\frac{2}{3}} = k\left[1 - \left(1 - \frac{2}{3}\frac{1-\epsilon}{k} - \frac{2}{3}\frac{1}{3}\frac{1}{2!}\frac{(1-\epsilon)^2}{k^2} - \cdots\right)\right] \geq \frac{2}{3}(1-\epsilon). \tag{21}$$

We also note that $\beta^{2k/3} = \left(1 - \frac{1-\epsilon}{k}\right)^{2k/3}$ is an increasing function of $k$, bounded above by $e^{2\frac{2}{3}(1-\epsilon)}$. Hence, $c_2(k) = (1 - \beta^{2k/3})$ is a decreasing function of $k$. We note that $k = 3$, $c_2(3) = 1 - \beta^{2k/3} = 1 - \beta^2 = (1 - 1 - \frac{(1-\epsilon)^2}{9} + \frac{2}{3}(1-\epsilon)) \leq \frac{2}{3}(1-\epsilon)$. Thus, we have shown that for $k \geq 3$,

$$c_2(k) \leq c_2(3) \leq \frac{2}{3}(1-\epsilon). \tag{22}$$

Using (21) and (22), we have that for $k \geq 3$, $c_1(k) \leq 1 \Rightarrow c(k) \leq k^2$.

- For $k = 1$, $\beta = \epsilon$ and we get $c(1) = 1 = k^2$.

- For $k = 2$, $\beta = (1+\epsilon)/2$ and we get $c_1(2) = \frac{1-\beta^{4/3}}{2(1-\beta^{2/3})} = \frac{1+\beta^{2/3}}{2} \leq 1 \Rightarrow c(2) \leq 2^2$.

## B.6 Algorithm for solving (9)

Using the substitution $t_i = (\mu - \log(\delta_i))\nu_i^{-2} > 0$, the problem (9) is equivalent to

$$\min_{t_i, \delta_i > 0 \forall i} \ (1-\epsilon) \sum_{i=1}^{k} \beta^{k-i} \sqrt{(\mu - \log(\delta_i))/t_i}$$
$$\text{s.t.} \ \sum_{i=1}^{k} t_i \leq c, \ \sum_{i=1}^{k} \delta_i \leq \delta. \tag{23}$$

Problem (23) is non-convex and we can resort to an iterative algorithm scheme to empirically solve it. In particular, we make use of the manifold optimization framework (Boumal, 2023) on the multinomial manifold (Sun et al., 2016) using the Manopt toolbox (Boumal et al., 2014).

## B.7 Proof of Theorem 3.9

The proof strategy of this result is similar to that employed for Theorem 3.5.

We require solving Problem (9) only with respect to $\{\nu_i\}_{i=1}^{k}$ and fix $\delta_i = \delta/k \ \forall i$. For this, we first apply the following substitution:

$$x_i = \frac{1}{\nu_i}\sqrt{\frac{b}{N_0}}, \tag{24}$$

where $b = \frac{18n}{k}\log(\frac{k}{\delta})\log(\frac{1}{\epsilon})$ for BSG-ABA and $b = \frac{2n}{k}\log(\frac{n}{\delta}\log(\frac{1}{\epsilon}))\log(\frac{1}{\epsilon})$ for BSG-NE.

Following this substitution, Problem (9) with $\delta_i = \delta/k \ \forall i$ can equivalently rewritten as

$$\min_{x_i > 0 \forall i} \ (1-\epsilon)\sqrt{\frac{b}{N_0}} \sum_{i=1}^{k} \frac{\beta^{k-i}}{x_i}$$
$$\text{s.t.} \ \sum_{i=1}^{k} x_i^2 \leq 1 \tag{25}$$

We now employ Lemma B.2 to solve the above Problem (25). The expression of optimal $\nu_i^*$ is

$$\nu_i^* = \frac{1}{x_i^*}\sqrt{\frac{b}{N_0}}, \tag{26}$$

where $x_i^*$ is given by (16), $a_i^2 = (1-\epsilon)\beta^{k-i}\sqrt{\frac{b}{N_0}}$, $r = 2$, and $p = 3/2$. The overall $\nu^*$ for BSG algorithms is computed as $\nu^* = (1-\epsilon)\sum_{i=1}^k \beta^{k-i}\nu_i^*$.

## C  Theorems and proofs related to the proposed BG algorithm

In this section, we re-state the theorems in the main paper for the proposed BG algorithm. The proofs follow the proof strategy of Section B.

**Lemma C.1.** *Given a current solution $\mathbf{s}$ of BG and let $(\nu_0, \delta_i)$ be the PAC parameters for the next iteration. Then, the following holds with probability $1 - \delta_i$: the gain of BG (Algorithm 1) in one iteration is at least $\frac{1}{k}\sum_{s\in\mathbf{s}^*\setminus\mathbf{s}}(g(s|\mathbf{s}) - \nu_0)$ where $\mathbf{s}^*$ denote the optimal solution of problem (2).*

*Proof.* The proof is similar to the proof of Lemma B.1 except that $\mathcal{R}$ is the entire set $\mathcal{S}\setminus\mathbf{s}$. In the given iteration, BG's step 2 (`PAC_BEST_ARM`) selects a $\nu_0$-optimal element $s$ with probability $1 - \delta_i$. That is,

$$g(s|\mathbf{s}) \geq \max_a g(a|\mathbf{s}) - \nu_0 \geq \frac{1}{|\mathbf{s}^*\setminus\mathbf{s}|}\sum_{s\in\mathbf{s}^*\setminus\mathbf{s}}(g(s|\mathbf{s}) - \nu_0) \geq \frac{1}{k}\sum_{s\in\mathbf{s}^*\setminus\mathbf{s}}(g(s|\mathbf{s}) - \nu_0).$$

with probability at least $1 - \delta_i$. This completes the poof of the lemma. $\qquad\square$

**Theorem C.2.** *Let $\mathbf{s}$ be the solution obtained by BG (BG-ABA or BG-NE) for (2) with given PAC parameters $(\nu, \delta)$. Then, $\mathbf{s}$ is a $(\nu, \delta)$-PAC solution of (2) if per-iteration $(\nu_i, \delta_i)$ policy is defined as*

$$\sum_{i=1}^k \delta_i = \delta \quad \text{and} \quad \sum_{i=1}^k \alpha^{k-i}\nu_i \leq \nu,$$

*where $\alpha = 1 - 1/k$.*

*Proof.* The proof follows the proof strategy of Theorem 3.1 and relies on Lemma C.1. The inequality corresponding to (13) is

$$A_{i+1} \leq \alpha A_i + \nu_{i+1},$$

where $A_{i+1} = F(\mathbf{s}^*) - F(\mathbf{s}_{i+1})$ and $\alpha = 1 - 1/k$. Summing it over from $i = 0$ to $i = k - 1$, we get $A_k \leq \alpha^k A_0 + \sum_{i=1}^k \alpha^{k-i}\nu_i$. Finally, we have the inequality $F(\mathbf{s}_k) \geq (1 - e^{-1})F(\mathbf{s}^*) - \nu$ with $\sum_{i=1}^k \alpha^{k-i}\nu_i \leq \nu$. $\quad\square$

**Lemma C.3.** *Let $N_{\mathrm{ABA}}$ and $N_{\mathrm{NE}}$ be the upper bounds on the number of stochastic function $f$ evaluations required by BG-ABA and BG-NE (Algorithm 1), respectively. Then,*

$$\begin{aligned} N_{\mathrm{ABA}} &= \textstyle\sum_{i=1}^k \frac{18n}{\nu_i^2}\log\left(\frac{1}{\delta_i}\right), \\ N_{\mathrm{NE}} &= \textstyle\sum_{i=1}^k \frac{2n}{\nu_i^2}\log\left(\frac{n}{\delta_i}\right). \end{aligned} \tag{27}$$

*Proof.* The proof is similar to the proof of Lemma 3.3 except $\mathcal{R}_i$ is all the remaining candidates set and $|\mathcal{R}_i|$ is $n - i$ whose upper bound is $n$ that is used in the expressions. $\qquad\square$

**Theorem C.4.** *Let $(\nu_i^*, \delta_i^*)$ be an optimal per-iteration policy corresponding to the minimum number of stochastic function $f$ calls required by BG to obtain a $(\nu, \delta)$-PAC solution. Then,*

$$\nu_i^* = \frac{\nu(1-\alpha^{2/3})\alpha^{(i-k)/3}}{(1-\alpha^{2k/3})}$$

*and $\delta_i^* = \delta/k$. The expressions of $N_{\mathrm{ABA}}$ and $N_{\mathrm{NE}}$ corresponding to $(\nu_i^*, \delta_i^*)$ are*

$$\begin{aligned} N_{\mathrm{ABA}}^* &= \frac{18nk}{\nu^2}\log\left(\frac{k}{\delta}\right)c(k) \text{ and} \\ N_{\mathrm{NE}}^* &= \frac{2nk}{\nu^2}\log\left(\frac{nk}{\delta}\right)c(k), \end{aligned}$$

*where $c(k) = \frac{(1-\alpha^{2k/3})^3}{k(1-\alpha^{2/3})^3}$ and $\alpha = 1 - 1/k$.*

*Proof.* We follow the proof of Theorem 3.5, make use Lemma B.2, and use the substitution

$$\sqrt{x_i} = \alpha^{k-i}\nu_i/\nu.$$

$\square$

**Theorem C.5.** *Given a fixed budget $N_0$ (the maximum number of stochastic function $f$ calls allowed) and confidence $\delta$, BG-ABA and BG-NE achieve $(\nu^*_{\mathrm{ABA}}, \delta)$- and $(\nu^*_{\mathrm{NE}}, \delta)$-PAC solutions, respectively, where*

$$\nu^*_{\mathrm{ABA}} = \left(\frac{1-\alpha^{2k/3}}{1-\alpha^{2/3}}\right)^{3/2}\left(\frac{18n\log(k/\delta)}{N_0}\right)^{1/2}$$
$$\nu^*_{\mathrm{NE}} = \left(\frac{1-\alpha^{2k/3}}{1-\alpha^{2/3}}\right)^{3/2}\left(\frac{2n\log(nk/\delta)}{N_0}\right)^{1/2}$$

*and $\alpha = 1 - 1/k$. The corresponding per-iteration policy of BSG-ABA is given by $\delta^*_i = \delta/k$ and*

$$\nu^*_i = \left(\frac{18n\log(k/\delta)(1-\alpha^{2k/3})}{N_0(1-\alpha^{2/3})}\right)^{1/2}\alpha^{(i-k)/3}.$$

*Similarly, the per-iteration policy of BSG-NE, corresponding to $(\nu^*_{\mathrm{NE}}, \delta)$, is given by $\delta^*_i = \delta/k$ and*

$$\nu^*_i = \left(\frac{2n\log(nk/\delta)(1-\alpha^{2k/3})}{N_0(1-\alpha^{2/3})}\right)^{1/2}\alpha^{(i-k)/3}.$$

*Proof.* We need to use Lemma B.2 after doing the following substitution:

$$x_i = \frac{1}{\nu_i}\sqrt{\frac{b}{N_0}},$$

where $b = 18n\log(\frac{k}{\delta})$ for BG-ABA and $b = 2n\log(\frac{nk}{\delta})$ for BG-NE. The above substitution implies $a_i^2 = \alpha^{k-i}\sqrt{\frac{b}{N_0}}$ in Lemma B.2. $\square$

# D   Empirical results on optimizing both $\{\nu_i\}_{i=1}^k$ and $\{\delta_i\}_{i=1}^k$ for learning policies

**Minimizing computational cost setting.**   Here, we compare the solution obtained by optimizing problem (8) in two settings: (Policy 1) the policy learned by having uniform per-iteration confidence, as discussed in Theorem 3.5, and (Policy 2) the policy learned by solving (8) via alternating minimization algorithm discussed in Appendix B.3.

Figure 3 shows the values and the trade-offs obtained by the two policies. Although the alternating minimization approach leads to a smaller computational cost (in terms of the number of stochastic function calls) than the one proposed in Theorem 3.5, the improvements are only around 0.23%.

**Fixed computational cost setting.**   Here, we compare the solution obtained by optimizing problem (9) in two settings: (Policy 1) the policy learned by having uniform per-iteration confidence, as discussed in Theorem 3.9, and (Policy 2) the policy learned by solving (9) via Riemannian gradient descent based algorithm discussed in Appendix B.6.

Figure 4 shows the values of $\{\nu_i\}$ and $\{\delta_i\}$ obtained by solving the problem (9). Although we obtain a smaller $\nu = (1 - \epsilon)\sum_{i=1}^k \beta^{k-i}\nu_i$ using joint optimization of $\{\nu_i\}$ and $\{\delta_i\}$ than that from Theorem 3.9, the improvement is around 0.17% for the problem setting in Figure 4.

In the above setting, we are given the confidence parameter $\delta$ along with the cost $c$. However, it might be interesting to look at how the optimized $\nu$ (i.e., the argmin solution to (9)) varies with different choices of $\delta$ but fixed $c$. Figure 5 shows the $(\nu, \delta)$ Pareto optimal curve for a given cost $c$. Here, $\delta \in \{10^{-8}, 10^{-7}, 10^{-6}, 10^{-5}, 10^{-4}, 10^{-3}, 10^{-2}, 10^{-1}, 0.2, 0.3\}$.

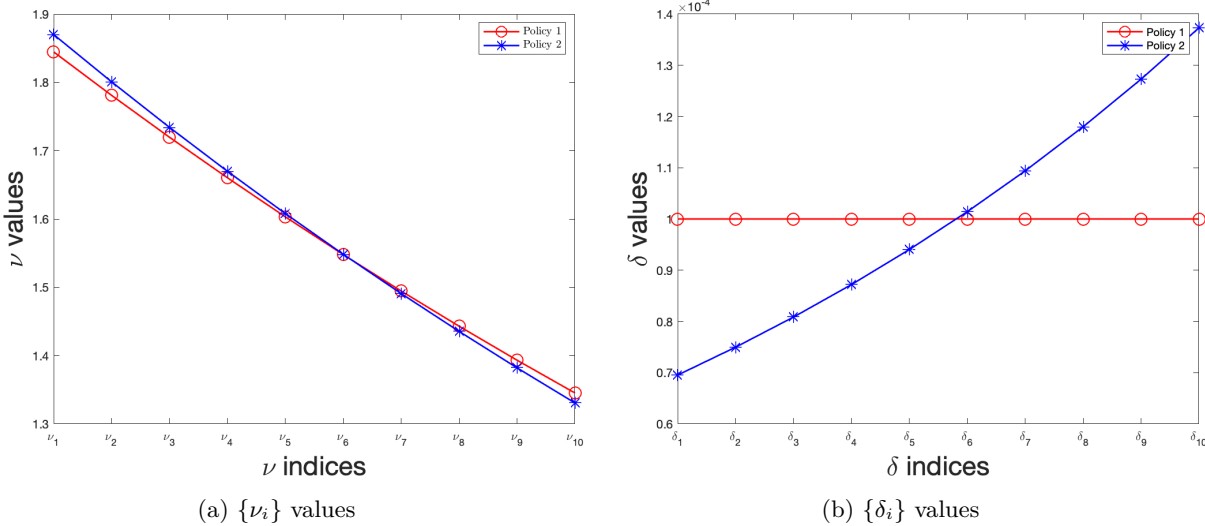

(a) $\{\nu_i\}$ values

(b) $\{\delta_i\}$ values

Figure 3: We plot the values of $\{\nu_i\}$ and $\{\delta_i\}$ for two different policies in the fixed $(\nu, \delta)$ setting. The problem instance has $k = 10$, $\nu = 10$, $\delta = 10^{-3}$, and $\epsilon = 10^{-3}$. Policy 1 refers to the values proposed in Theorem 3.5 and $\delta_i = \delta/k = 10^{-4}$. Policy 2 refers to the solutions obtained by solving the problem (8) which optimizes both $\{\nu_i\}$ and $\{\delta_i\}$.

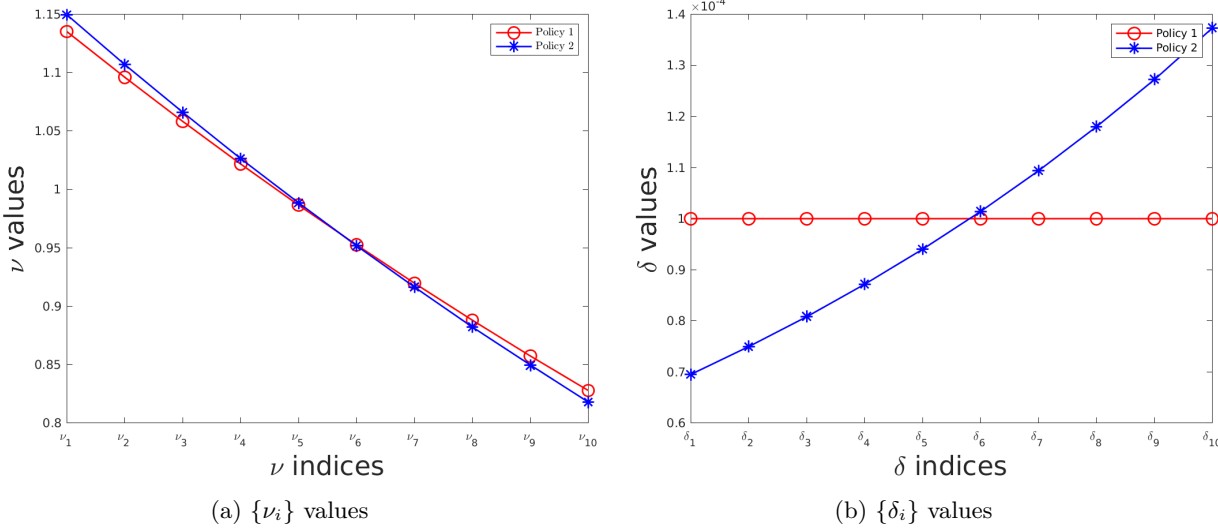

(a) $\{\nu_i\}$ values

(b) $\{\delta_i\}$ values

Figure 4: We plot the values of $\{\nu_i\}$ and $\{\delta_i\}$ for two different policies in the fixed cost setting. The problem instance has $k = 10$, $\delta = 10^{-3}$, $\epsilon = 10^{-3}$, and $c = 100$. Policy 1 refers to the values proposed in Theorem 3.9 and $\delta_i = \delta/k = 10^{-4}$. Policy 2 refers to the solutions obtained by solving the problem (9) which optimizes both $\{\nu_i\}$ and $\{\delta_i\}$.

# E   Additional discussion

In this section, we discuss a few other details related to the developments in the main paper.

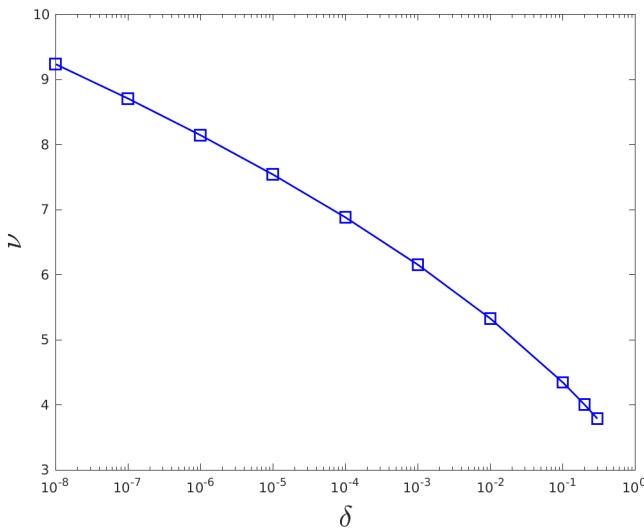

Figure 5: Pareto optimal curve in the fixed cost setting.

### E.1 On ProtoBandit (Roy Chaudhuri et al., 2022)

The recent work of Roy Chaudhuri et al. (2022) also uses a MAB setting to solve exemplar-based clustering problem. In particular, they also makes use of the ABA algorithm. However, a number of crucial differences exist that are highlighted below.

- The focus in (Roy Chaudhuri et al., 2022) is limited to exemplar-based clustering problem. On the other hand, we have proposed results for general SSM problems of type (2).

- Roy Chaudhuri et al. (2022) do not discuss optimizing the $(\nu_i, \delta_i)$ policy and use $\nu_i = \nu_0$ (i.e., fixed user-defined $\nu_i$) at every iteration of ProtoBandit. In contrast, we propose how to optimize the per-iteration $(\nu_i, \delta_i)$ policies in different settings.

- For the exemplar-based clustering problem, the number of pairwise distance computations required by ProtoBandit is $O(nk^3)$, whereas our proposed BSG-ABA algorithm require $O(nk^2)$ pairwise distance computations. Hence, our algorithms show an improvement of a factor $k$.

- As detailed in (Roy Chaudhuri et al., 2022, Section 3.5), they implement a KL-LUCB algorithm based method with early stopping heuristic for empirical study. However, they do not provide any PAC style analysis of their implemented algorithm.

### E.2 Regarding Remark 3.8 on BanditPAM (Tiwari et al., 2020)

For BanditPAM (Tiwari et al., 2020) and the proposed algorithms, the confidence $\delta$ mentioned in our draft corresponds to the $1 - \delta$ confidence of the PC guarantee (BanditPAM) and the $1 - \delta$ confidence of the PAC guarantee (proposed algorithms).

The BanditPAM paper mentions a different $\delta$ parameter, henceforth referred to as $\delta_{\mathrm{BP}}$. The $\delta_{\mathrm{BP}}$ parameter in BanditPAM's draft (and code) corresponds to $1 - 2n^2\delta_{\mathrm{BP}}$ confidence for every BanditPAM iteration. Tiwari et al. (2020, Remark A1 in the Appendix section) details this. Hence, in terms of the $\delta_{\mathrm{BP}}$ parameter, BanditPAM guarantees the following across $k$ iterations. Given the parameter $\delta_{\mathrm{BP}}$, Tiwari et al. (2020) provide the following guarantee: With probability at least $1 - 2n^2k\delta_{\mathrm{BP}}$, BanditPAM (restricted to the submodular BUILD step) returns the same set of $k$ points as Greedy and the required number of stochastic function $f$ calls ($N_{\mathrm{BP}}$) satisfies $\mathbb{E}[N_{\mathrm{BP}}] = O(n^4 d_1(k)\delta_{\mathrm{BP}} + nd_2(k)\log(1/\delta_{\mathrm{BP}}))$.

---

**Algorithm 2** Approximate best arm (ABA) algorithm

---

1: **Input: s**, $\mathcal{R}$, $\mathcal{Z}$, $\nu, \delta$
2: $\mathbf{t} = \mathcal{R}$.
3: **if** $n < \max\{10^5, \delta^{-4}\}$ **then**
4:    $s \leftarrow \texttt{naiveElimination}(\mathbf{s}, \mathbf{t}, \mathcal{Z}, \nu, \delta)$
5: **else**
6:    $\mathbf{r} \leftarrow \frac{|\mathbf{t}|^{7/8}}{2}$ elements selected uniformly at random from set $\mathcal{R}$.
7:    $\mathbf{t}_1 \leftarrow \texttt{aggressiveElimination}(\mathbf{s}, \mathbf{t}, \mathcal{Z}, (1 - 1/e)\nu, \frac{\delta}{2})$
8:    $s \leftarrow \texttt{naiveElimination}(\mathbf{s}, \mathbf{t}_1 \cup \mathbf{r}, \mathcal{Z}, \frac{\nu}{e}, \frac{\delta}{e})$
9: **end if**
10: **Output:** $s$. $\{s \text{ is a } (\nu, \delta)\text{-best element}\}$

---

### E.3 On (Karimi et al., 2017, Proposition 3)

We begin by noting that Karimi et al. (2017) have proposed a multilinear extension based approach for maximizing weighted coverage functions, which is a subclass of the SSM problem (2). As discussed in (Karimi et al., 2017, Section 3), the multilinear extension $G$ of a submodular function $F$ lacks concavity guarantee and may suffer from bad local optima. Thus, they consider concave continuous extensions of $G$ that are efficient to compute and are at most a constant factor away from $G$ to ensure a good quality solution. Weighted coverage functions is one such class of (stochastic) submodular functions where such an extension can be efficiently computed.

For weighted coverage functions, the stochastic function $f$ is submodular. However, as noted in (Mokhtari et al., 2018; 2020), the stochastic function $f$ need not be submodular for SSM problems. Overall, Karimi et al. (2017) propose a concave relaxation scheme and employed projected stochastic gradient ascent (SGA) algorithm. For the SSM problem (2), SGA does not provide tight guarantees as it offers $\text{OPT}/2 - \nu$ lower bound in expectation after $O(n^2k^2/\nu^2)$ iterations (Hassani et al., 2017).

Karimi et al. (2017) have also analyzed a vanilla Hoeffding bound based strategy for maximizing weighted coverage functions. Karimi et al. (2017, Proposition 3) crucially assume the submodularity of the stochastic function $f$. As discussed above, this assumption is not made in the subsequent SSM literature (Mokhtari et al., 2018; 2020; Hassani et al., 2019; 2020). Following these later works on SSM, we propose an approach for the general stochastic submodular function $F$ in (2) and do not assume submodularity of the stochastic function $f$.

## F Experiment details

Some additional details regarding experimental settings are below.

- For BanditPAM, we use the code from `https://github.com/motiwari/BanditPAM/pull/262` which fixes a critical bug (`https://github.com/motiwari/BanditPAM/issues/252`). In addition, we empirically do not observe much difference in BanditPAM's results with different values of the confidence parameter $\delta = \{0.01, 0.001, 2k/n\}$. In this last setting, the $\delta$ varies with $k$.

- For SCG, the error parameter $\nu$ influences only the number of iterations. In the implementation, we lower bound the number of SCG iterations with 1000 and upper bound it by 20000. With 20000 iterations, SCG takes around 42 minutes on the TIN dataset for $k \geq 90$ and is the slowest baseline in terms of time). Since the number of iterations for SCG with different $\nu$ parameters (at high $k$ values) is the same (20000), their plots coincide.

## G Bandit algorithms for PAC_BEST_ARM in Algorithm 1

We detail the ABA algorithm (Hassidim et al., 2020) in Algorithm 2. It employs the naive elimination and aggressive elimination strategies, which are detailed in Algorithms 3 and 4, respectively.

---

**Algorithm 3** Naive elimination algorithm

---

1: **Input: s**, $\mathcal{R}$, $\mathcal{Z}$, $\nu, \delta$
2: $\mathbf{t} = \mathcal{R}$.
3: Sample a subset $\mathcal{Z}_l \subseteq \mathcal{Z}$ with $l = \left\lceil \frac{2}{\nu^2} \log \frac{|\mathbf{t}|}{\delta} \right\rceil$ samples.
4: $s = \underset{a \in \mathbf{t}}{\arg\max}\, h(a|\mathbf{s}; \mathcal{Z}_l)$.
5: **Output:** $s$. $\{s$ is a $(\nu, \delta)$-best element$\}$

---

---

**Algorithm 4** Aggressive elimination algorithm

---

1: **Input: s**, $\mathcal{R}$, $\mathcal{Z}$, $\nu, \delta$
2: $\mathbf{t} = \mathcal{R}$.
3: $\mathbf{t}_0 = \mathbf{t}, t = |\mathbf{t}_0|, \phi(t) = \sqrt{\frac{6 \log t}{t^{3/4}}}, T(t) = \left\lceil \frac{\log t + 4 \log 2}{4 \log \frac{1}{\delta + \phi(t)}} \right\rceil$.
4: **for** $\ell \in \{0, 1, 2, \ldots, T(t)\}$ **do**
5:     Sample a subset $\mathcal{Z}_l \subseteq \mathcal{Z}$ with $l = (\ell + 1) \left\lceil \frac{2}{\nu^2} \log \frac{1}{\delta} \right\rceil$ samples .
6:     For each element $a \in \mathbf{t}_\ell$, compute $h(a|\mathbf{s}; \mathcal{Z}_l)$ and sort the elements in descending order based on the value of $h(a|\mathbf{s}; \mathcal{Z}_l)$
7:     $\mathbf{t}_{\ell+1} \leftarrow$ the top-$(|\mathbf{t}_\ell| \lfloor \delta + \phi(|t|) \rfloor)$ elements in $\mathbf{t}_\ell$.
8: **end for**
9: **Output:** the set of elements $\mathbf{t}_{T(t)+1}$.

---

