# OpenReview forum: "Revisiting stochastic submodular maximization with cardinality constraint: A bandit perspective"
_TMLR — Accepted by TMLR_

### Review · Reviewer_CzSb · 2023-12-02

**Summary Of Contributions:**

This paper focuses on the problem of maximizing non-negative, monotone, stochastic submodular functions under cardinality constraint. The paper considers a discrete, (stochastic) greedy setting under a probably approximately correct (PAC) setting, i.e., the goal is to obtain solutions whose expected objective value is greater than or equal to $(1-1/e-\varepsilon)OPT-\nu$ with at least $1-\delta$ probability.
The authors propose novel bandit stochastic greedy (BSG) algorithms, where the selection of the next element at iteration i is posed as a ($\nu_i$, $\delta_i$)-PAC best arm identification problem.

Given ($\nu$, $\delta$)-PAC parameters to BSG, the authors define a set of per-iteration ($\nu_i$, $\delta_i$)-policies such that any policy from this set guarantees a ($\nu$, $\delta$)-PAC solution for the stochastic submodular maximization problem using BSG. Next, they derive the optimal ($\nu^*_i$, $\delta^*_i$)-policy from this set which incurs the least computational cost in terms of the number of stochastic function calls ($N$). With the obtained optimal policy, they show that BSG has better complexity in $N$ than the existing approaches. Lastly, they further analyze the fixed-budget setting. Experiments illustrate the efficacy of the proposed approaches.

**Audience:**

Yes

**Broader Impact Concerns:**

I do not have ethical concerns.

**Claims And Evidence:**

Yes

**Requested Changes:**

Please see the review on weaknesses above.

**Strengths And Weaknesses:**

**Strengths:**

1.	The considered problem, stochastic submodular maximization with cardinality constraint, is important and finds many real-world applications, such as social influence maximization, exemplar-based clustering, recommender systems and prototype selection.
2.	The idea of regarding the selection of the next element at each iteration as a best arm identification problem is interesting and novel to the stochastic submodular maximization literature.
3.	The improvement of the number of stochastic function calls ($N$) over existing submodular maximization algorithms is significant, which makes a solid technical contribution to the literature.
4.	The authors also conduct experiments to validate the superiority of their algorithms in terms of optimization quality and computational efficiency.


**Weaknesses:**

1.	In table 1, it seems that the authors compare the number of f calls mainly with respect to $n$. The authors should give more discussion on other parameters, such as $m$ and $\nu$. How do the sizes of $n$, $m$ and $\nu$ influence the performance superiority of the compared algorithms?
2.	The authors should comment more on the theoretical results for the fixed-budget setting (Theorem 3.8) and connect these results to the literature. For example, it would be better to give a comparison between these results and existing fixed-budget results or any applicable lower bounds.

---

### Review · Reviewer_6Yj6 · 2023-12-18

**Summary Of Contributions:**

This paper considered the submodular maximization problem with a bandit-like algorithm. In detail, it proposed the bandit greedy (BG) and bandit stochastic greedy (BSG) algorithm, where an ABA or NE can be applied as a sub-routine. This work compared its performance to existing algorithms both theoretically and numerically.

**Audience:**

Yes

**Broader Impact Concerns:**

No ethical concern.

**Claims And Evidence:**

No

**Requested Changes:**

Please refer to the 'Weaknesses' part in the **Strengths And Weaknesses** section.

**Strengths And Weaknesses:**

Strengths:
1. The paper is overall easy to follow.
1. It is great to provide a table to compare the theoretical performance among algorithms in terms of both approximation guarantee and number of calls of function computation.

Weaknesses:
1. I would appreciate a better comparison between algorithms in Table 1.
    1. In the discrete setting, the Greedy algorithm provides a deterministic approximation guarantee while the BG algorithm provides a probabilistic one.
    1. In the stochastic setting, I think it may not be that difficult to extend the results to continuous bandit setting and provide a better comparison to existing algorithms.
1. Except for Section 3.4, the paper focused on the fixed-confidence setting of best arm identification. In the fixed confidence setting, the BG/BSG algorithm is proved to stop and identify a satisfactory set after a certain number of time steps. However, I doubt if such a fixed-confidence algorithm and analysis can be trivially applied as in Appendix B.5.
    1. First of all, Algorithm 1 may not terminate within a fixed time horizon with probability 1. Eq. 9 is not the correct definition of best arm identification in multi-armed bandits.
    1. The authors may consider to revise Algorithm 1 and the analysis for the fixed-budget setting. The theoretical guarantee should be verified.
1. In the experiment setting, I wonder what is the source of randomness. Why do we need the BSG algorithm? Why not simply the BG algorithm? Besides, what is the function $f$ in the experiment section?
1. Novelty: I am curious why the author(s) named Algorithm 1 as a bandit-type algorithm instead of a greedy algorithm? As online optimization problem has been studied for years, I suggest the author(s) to highlight their significant contributions or analytical challenges.

---

### Review · Reviewer_gLG2 · 2024-01-24

**Summary Of Contributions:**

The authors consider the problem of maximizing a submodular function $F(s)$ over the subsets of $[n]$ under the cardinality constraint $|s| \le k$. They further consider the case in which the evaluation of $F$ might be expensive, because it is given as an expectation $F(s) = E(f(z,s))$ where $z \in Z$ and has distribution $P$. Denote by $m$ the size of $Z$, in order to compute $F$ exactly one would have to enumerate the elements of $Z$ requiring $m$ evaluations,  which might be infeasible if $Z$ is large. Instead, the authors propose to use sampling strategies, in the sense that one may draw samples from $P$, and the goal is find an approximate maximizer of $F(s)$ with a limited number of samples.

The authors propose to modify the greedy algorithm for their setting, by running a best arm identification algorithm $k$ times, in order to mimic the way the greedy algorithm would select the successive items added to $s$.

**Audience:**

Yes

**Broader Impact Concerns:**

Not Applicable.

**Claims And Evidence:**

Yes

**Requested Changes:**

The authors should clarify how their approach compares with the simple approach highlighted above.

**Strengths And Weaknesses:**

I might have missed something, but I believe that the solution proposed by the authors is not better than the following simple solution:

- Choose $u = {k \over \nu^2} \log({2 n \over \delta}) $ samples drawn i.i.d. from $P$, and denote them by $Z_1,...,Z_u$
- Define the function $\Phi_u(s) = {1 \over u} \sum_{i=1}^u f(s,Z_i)$
- Maximize $\Phi_u(s)$ over $s$ under the constraint that $|s| \le k$ using the basic greedy algorithm

One can readily check that evaluating $\Phi_u(s)$ for a given value of $s$ requires $u$ functions evaluations, so that the greedy algorithm will require at most $O(n k u) = O(n {k^2 \over \nu^2} \log({n \over \delta}) )$ function evaluations in total.

Furthermore, with probability greater than $1-\delta$ we have that
$$\max_{s: |s| \le k} |\Phi_u(s) - F(s)| \le \nu$$
so that maximizing $\Phi_u$ will yield the same result as maximizing $F$, up to an additive error of $\nu$.

Indeed, $\Phi_u(s)$ is an i.i.d. average of bounded random variables in $[0,1]$ with expectation $F(s)$ so from Hoeffding's inequality
$$
P(  |\Phi_u(s) - F(s)| \ge \nu   ) \le 2 \exp(- 2 u \nu^2 )
$$
and since there are at most $n^k$ sets $s \subset [n]$ verifying the constraint $|s| \le k$,  a union bound yields
$$
P( \max_{s: |s| \le k} |\Phi_u(s) - F(s)| \ge \nu   ) \le 2 n^k \exp(- 2 u \nu^2 )
$$
Letting $u = {k \over \nu^2} \log({2 n \over \delta})$ proves that indeed:
$$
P( \max_{s: |s| \le k} |\Phi_u(s) - F(s)| \ge \nu   ) \le \delta
$$

So this trivial approach yields the same guarantees as that of the authors: approximation guarantee is $F(s) \ge \gamma(0) OPT - \nu$ with probability greater than $1-\delta$ and number of function evaluations is $O(n {k^2 \over \nu^2} \log({n \over \delta}) )$.

---

> ### Author Response · Authors · 2024-01-26
> **Response to Reviewer gLG2**
>
> In the solution given by the reviewer, one would be required to assume that the stochastic functions $f(s, Z)$ are submodular over the set $s$ for each realization of the random value of $Z = Z_i$. However, in our work, we do not make such a presumption and only require that $F(s) = \mathbb{E}_{Z \sim P}[f(s,Z)]$ (defined as expectation over $Z$) is submodular. The need for this stronger assumption in the reviewer's solution stems for the following reason.
>
> Let $OPT = \max_{s:|s|\leq k} F(s)$, $\hat{s}\_k^{*}=\arg\max_{s:|s|\leq k} \Phi_u(s)$, and $\hat{s}\_k$ be the solution obtained by the greedy algorithm while solving $\max_{s:|s|\leq k} \Phi_u(s)$. The  proof presented by the reviewer provides the following result
> $\max_{s:|s|\leq k} |\Phi_u(s)-F(s)| \leq \nu \mbox{  w.p. } 1-\delta$ when $u = \frac{k}{\nu^2} \log\left(\frac{2n}{\delta}\right)$ samples of $Z$ are drawn independently from $P$. This implies
>
> $OPT - \nu \leq \Phi\_u(\hat{s}\_k^{*})$ w.p. $1-\delta$. **(A)**
>
> However, we need a bound between $OPT$ and $\Phi_u(\hat{s}_k)$. This can be obtained if one assumes that the functions $f(s,Z_i)$ are submodular. This assumption implies that the function $\Phi_u(s)$ is submodular and hence the greedy approximation guarantee for submodular maximization can be applied:
>
> $(1-1/e)\Phi_u(\hat{s}_k^{*}) \leq \Phi_u(\hat{s}_k)$.  **(B)**
>
>
> Using the result **(B)** in **(A)** gives us the required bound in the reviewer's proof, but only when the submodularity of the functions $f(s,Z_i)$ is assumed.
>
> Following existing works (Mokhtari et al., 2018; 2020), we do not assume that the functions $f(s,Z_i)$ are submodular. This is also mentioned in the sentence below equation (2). Hence, we do not approximate the stochastic submodular function $F(s)$ via a function $\Phi(\cdot)$. Instead, as discussed in Section 3.1, we propose to learn a $(\nu_i,\delta_i)$ probably approximately correct solution for the incremental gain problem at every iteration $i$ of the proposed bandit stochastic greedy (BSG) algorithm. While this provides a PAC-guarantee for each iteration, how does it relate to the overall PAC-guarantee of BSG in the context of maximizing $F(s)$? Which per-iteration $\{(\nu_i,\delta_i)\}_{i=1}^k$ policies correspond to the lowest computational cost or the fixed-budget settings? These questions are subsequently discussed and answered in Sections 3.2-3.4.

---

> ### Comment · Reviewer_gLG2 · 2024-01-29
>
> In fact, even if one assumes that the mapping from $s$ to $f(s,z)$ is not submodular, I believe that my previous remark still holds. The only important condition is that the mapping from $s$ to $F(s)$ is submodular.
>
> This is true from a (very minor) modification of the analysis of Nemhauser, reproduced in the survey of Krause and Golovin "Submodular Function Maximization". For clarity, we will use their notation.
>
> Indeed, define $s^i$ the output of the greedy algorithm applied to $\Phi_u$ under the cardinality constraint $|s| \le i$ define $s^\star = \\{v_1^\star,...,v_k^\star\\}$ the maximizer of $F(s)$ subject to $|s| \le k$ and define $\Delta_{F}(e|s) = F(s \cup e) - F(s)$ the discrete derivative.
>
> Since $F$ is monotone:
> $$
> 	F(s^\star) \le F(s^\star \cup s^i)
> $$
> Writing the above as the sum of its discrete derivatives
> $$
> 	F(s^\star \cup s^i) = \sum_{j=1}^k \Delta_{F}(v_j^\star | s^i \cup \\{v_1^\star,...,v_{j-1}^\star\\} )
> $$
> Since $F$ is submodular its discrete derivative is a decreasing function
> $$
> 	\sum_{j=1}^k \Delta_{F}(v_j^\star | s^i \cup \\{v_1^\star,...,v_{j-1}^\star\\} ) \le \sum_{v \in s^\star} \Delta_{F}(v | s^i)
> $$
> Since $s^{i+1}$ is the output of the greedy algorithm applied to $\Phi_u$, it is the maximizer of $\Phi_u(v \cup s^i)-\Phi_u(s^i)$, and since $\max_{s: |s| \le k} |F(s) - \Phi_u(s) | \le \nu$ we have:
> $$
> 	\Delta_{F}(v | s^i) \le \max_{v} \Delta_{F}(v | s^i) \le 2 \nu + F(s^{i+1}) - F(s^{i})
> $$
> Putting everything together we have proven
> $$
> 	F(s^\star) \le F(s^i) + k (F(s^{i+1}) - F(s^i)) + 2 k \nu
> $$
> Letting $\delta_i= F(s^\star) - F(s^i) - 2 k \nu$ the above can be rewritten as:
> $$
> \delta_i \le k(\delta_{i+1} - \delta_{i})
> $$
> hence since $F$ is bounded by $1$:
> $$
> \delta_{k} \le \delta_{0} (1 - {1 \over k})^k \le {1 \over e}
> $$
> This gives the approximation guarantee:
> $$
> 	F(s^k) \ge F(s^\star) (1-1/e) - 2 k \nu
> $$
> The analysis above can be found verbatim (when $\nu=0$), in the survey of Krause and Golovin "Submodular Function Maximization".

---

> > ### Author Response · Authors · 2024-01-30
> > **Additional response to Reviewer gLG2**
> >
> > We thank the reviewer for sharing additional details of the solution. In the proof shared by the reviewer, it is shown that one obtains the approximation guarantee
> > $$
> > F(s^k) \ge F(s^\star) (1-1/e) - 2 k \nu_0
> > $$
> > with $O(nk^2\log(\frac{n}{\delta})/\nu_0^2)$ number of function $f$ evaluations.
> > By taking $\nu=2k\nu_0$, this is equivalent to the approximation guarantee of
> > $$
> > F(s^k) \ge F(s^\star) (1-1/e) - \nu
> > $$
> > with $O(4nk^4\log(\frac{n}{\delta})/\nu^2)$ number of function $f$.
> > So the number of function $f$ evaluations using this strategy is **more** than the proposed approach.
> >
> > A key difference between the reviewer's suggested approach and the proposed approach is as follows. The reviewer's suggested approach essentially considers $n^k$ arms (for all subsets $s$ with $|s| \leq k$) and sample accordingly ($\sim O(\log(n^k))$) to obtain an approximation $\Phi_u(s)$ of the function $F(s)$. The function $\Phi_u(s)$ is subsequently maximized iteratively using the greedy selection. On the other hand, we consider a maximum of $n$ arms in each greedy iteration and sample accordingly ($\sim O(\log(n)$) to approximately solve (the simpler) the maximum incremental gain problem. Hence, both the approximation and the greedy selection is done jointly in our proposed approach, and overall, our proposed approach incurs lower computational cost.

---

### Decision · Action_Editor_5pJy · 2024-03-05

**Recommendation:** Accept with minor revision

**Comment:**

The reviewers generally agreed that the paper satisfies the criteria of TMLR. The authors adequately addressed reviewer comments. See my comments about claims and evidence for minor concerns that should be addressed in the camera ready.

**Audience:**

This paper would be of interest to those working on submodular optimization and bandit problems.

**Claims And Evidence:**

The paper proposes a new method for stochastic submodular optimization. The paper makes a few key claims:
- it finds the set of per-iteration $(\nu_i,\delta_i)$ policies that lead to a $(\nu,\delta)$-PAC solution for BSG
- it finds the optimal policy in terms of calls to the function $f$ for a fixed $(\nu,\delta)$
- it finds the lowest error policy for a fixed function call budget

Although I think the paper satisfies the criteria for TMLR, I do have issues with the evidence presented for all of these claims. I believe they can and should be addressed in a minor revision.
- Q1 on p2 asks "which set of $(\nu_i, \delta_i)$ policies guarantee a $(\nu,\delta)$ PAC solution for BSG"? If I understand correctly, your answer to that question is Theorem 3.1. But Theorem 3.1 does not establish *the* set of $(\nu_i, \delta_i)$ policies; it establishes a *subset* of such policies. Either the text needs to be edited to change the claim, or the theorem needs to be fixed to be "if and only if"
- Q2 on p2 asks "what is the optimal policy in terms of function calls?". Again, it appears Theorem 3.4 is the answer to this question. But Theorem 3.4 establishes the optimal policy within $A(\nu, \delta)$ (and only for fixed $\delta_i = \delta/k$), not the overall optimal policy (unless Theorem 3.1 is amended to show that $A(\nu,\delta)$ is the entire set of valid PAC policies).
- Same general point about Q3 as I made for Q2 and Q1.
- The statements of many theoretical results in the paper are awkward / unclear. Many use the form "Let ... be a bound/optimal policy/etc. Then [display] .... = .... ." But this can lead to unclear statements. One particularly bad example is Lemma 3.3, which says (I'm summarizing here): "Let $N$ be upper bounds on $f$ evals. Then $N = \dots$." It's not clear what "Let $N$ be upper bounds" means; it sounds like it's defining $N$, but the definition is ambiguous. I would expect the lemma to just read something like "The number of evaluations $N$ satisfies [display] $N \leq \dots$. Furthermore, this bound is tight." or some such. There are other theoretical results that use this pattern leading to unclear statements.
- It's not clear what "tight" means in lemma 3.3. In the proof, it appears the paper simply shows that the case where $k=1$ has equality. This shows the bounds are tight only when $k=1$; for $k>1$ they may be loose. Is there even a reason to assert tightness here?